# Network inference from glycoproteomics data reveals new reactions in the IgG glycosylation pathway

Elisa Benedetti[1], Maja Pučić-Baković[2], Toma Keser [3], Annika Wahl[4,5], Antti Hassinen[6], Jeong-Yeh Yang[7], Lin Liu[7], Irena Trbojević-Akmačić[2], Genadij Razdorov [2], Jerko Štambuk[2], Lucija Klarić[2,8,9], Ivo Ugrina [3,10,11], Maurice H.J. Selman[12], Manfred Wuhrer [12], Igor Rudan [8], Ozren Polasek[13,14], Caroline Hayward [9], Harald Grallert[4,5,15], Konstantin Strauch[16,17], Annette Peters[5], Thomas Meitinger[18], Christian Gieger[4,5], Marija Vilaj[2], Geert-Jan Boons[7,19], Kelley W. Moremen[7], Tatiana Ovchinnikova[20], Nicolai Bovin[20], Sakari Kellokumpu[6], Fabian J. Theis [1,21], Gordan Lauc[2,3] & Jan Krumsiek [1,15]

Immunoglobulin G (IgG) is a major effector molecule of the human immune response, and aberrations in IgG glycosylation are linked to various diseases. However, the molecular mechanisms underlying protein glycosylation are still poorly understood. We present a data-driven approach to infer reactions in the IgG glycosylation pathway using large-scale mass-spectrometry measurements. Gaussian graphical models are used to construct association networks from four cohorts. We find that glycan pairs with high partial correlations represent enzymatic reactions in the known glycosylation pathway, and then predict new biochemical reactions using a rule-based approach. Validation is performed using data from a GWAS and results from three in vitro experiments. We show that one predicted reaction is enzymatically feasible and that one rejected reaction does not occur in vitro. Moreover, in contrast to previous knowledge, enzymes involved in our predictions colocalize in the Golgi of two cell lines, further confirming the in silico predictions.

[1] Institute of Computational Biology, Helmholtz Zentrum München—German Research Center for Environmental Health, 85764 Neuherberg, Germany. [2] Genos Glycoscience Research Laboratory, 10000 Zagreb, Croatia. [3] Faculty of Pharmacy and Biochemistry, University of Zagreb, 10000 Zagreb, Croatia. [4] Institute of Epidemiology 2, Research Unit Molecular Epidemiology, Helmholtz Zentrum München—German Research Center for Environmental Health, 85764 Neuherberg, Germany. [5] Institute of Epidemiology 2, Helmholtz Zentrum München—German Research Center for Environmental Health, 85764 Neuherberg, Germany. [6] Faculty of Biochemistry and Molecular Medicine, University of Oulu, FI-90014 Oulu, Finland. [7] Complex Carbohydrate Research Center, University of Georgia, Athens, GA 30602, USA. [8] Usher Institute of Population Health Sciences and Informatics, University of Edinburgh, EH8 9AG Edinburgh, UK. [9] Medical Research Council Human Genetics Unit, Institute of Genetics and Molecular Medicine, University of Edinburgh, EH4 2XU Edinburgh, UK. [10] Faculty of Science, University of Split, 21000 Split, Croatia. [11] Intellomics Ltd., 10000 Zagreb, Croatia. [12] Leiden University Medical Center, 2333 ZA Leiden, The Netherlands. [13] University of Split School of Medicine, 21000 Split, Croatia. [14] Gen-info Ltd., 10000 Zagreb, Croatia. [15] German Center for Diabetes Research (DZD), 40225 Düsseldorf, Germany. [16] Institute of Genetic Epidemiology, Helmholtz Zentrum München—German Research Center for Environmental Health, 85764 Neuherberg, Germany. [17] Institute of Medical Informatics, Biometry and Epidemiology, Chair of Genetic Epidemiology, Ludwig-Maximilians Universität, 81577 Munich, Germany. [18] Institute of Human Genetics, Helmholtz Zentrum München—German Research Center for Environmental Health, 85764 Neuherberg, Germany. [19] Department of Chemical Biology and Drug Discovery, Utrecht Institute for Pharmaceutical Sciences, and Bijvoet Center for Biomolecular Research, Utrecht University, 3584 CG Utrecht, The Netherlands. [20] Shemyakin and Ovchinnikov Institute of Bioorganic Chemistry, Russian Academy of Sciences, 117997 Moscow, Russia. [21] Department of Mathematics, Technical University Munich, 85748 Garching bei München, Germany. Correspondence and requests for materials should be addressed to F.J.T. (email: fabian.theis@helmholtz-muenchen.de) or to G.L. (email: glauc@genos.hr) or to J.K. (email: jan.krumsiek@helmholtz-muenchen.de)

Most membrane and secreted proteins are glycosylated, giving the information flow in biological systems an additional layer of complexity[1]. Immunoglobulin G (IgG) is responsible for the majority of antibody-based immunity in humans and is the most abundant glycoprotein in blood[2]. Like all antibodies, soluble IgG is produced and secreted by B lymphocytes and has two functional domains: an antigen-binding fragment (Fab), which is responsible for recognizing antigens on foreign pathogens and infected cells and a crystallizable fragment (Fc), which triggers the immune response by interacting with various Fc receptors[3]. The Fc domain contains a highly conserved glycosylation site at asparagine 297[4], to which a variety of glycan structures can be attached. Alternative Fc glycosylation alters the affinity of IgG to virtually all Fc receptors[5,6] and therefore plays an essential role in mediating the immune response[3,7]. Furthermore, aberrant glycosylation has been linked to various diseases, including rheumatoid arthritis[8], diabetes[9], and cancer[10]. Therefore, there is a need to elucidate how IgG glycans are synthesized and regulated in order to better understand their involvement in the human antibody-based immune response.

Current knowledge about the protein glycosylation pathway is likely to be incomplete, as our understanding of the complex glycan biosynthesis pathway is based solely on in vitro experiments, which have established the substrate specificity of major glycosyltransferase enzymes[11]. Unfortunately, due to the complexity of the glycosylation process, the in vivo experimental validation that is required to account for intracellular localization and protein-specific and site-specific glycosylation is still unfeasible, and currently available measurement techniques do not allow glycosylation to be analyzed at a subcellular level, making it impossible to experimentally verify whether a given glycosylation reaction that is enzymatically possible in vitro actually occurs in the cell. Thus, gaining a more precise picture of protein glycosylation at the molecular level would further our understanding of how the process is regulated in vivo and possibly identify key elements that alter glycan profiles during pathological processes. In case of IgG glycosylation, this is expected to guide the development of new pharmacological approaches that could replace cumbersome intravenous immunoglobulin therapy[12].

This study attempts to fill part of this knowledge gap using plasma IgG glycomics liquid chromatography-mass spectrometry (LC-MS) measurements from four independent cohorts to infer the enzymatic reactions that are involved in the IgG glycosylation pathway (Fig. 1). To do this, we first generate a partial correlation network, also known as a Gaussian graphical model (GGM). In the GGM, the nodes represent individual glycans and the edges

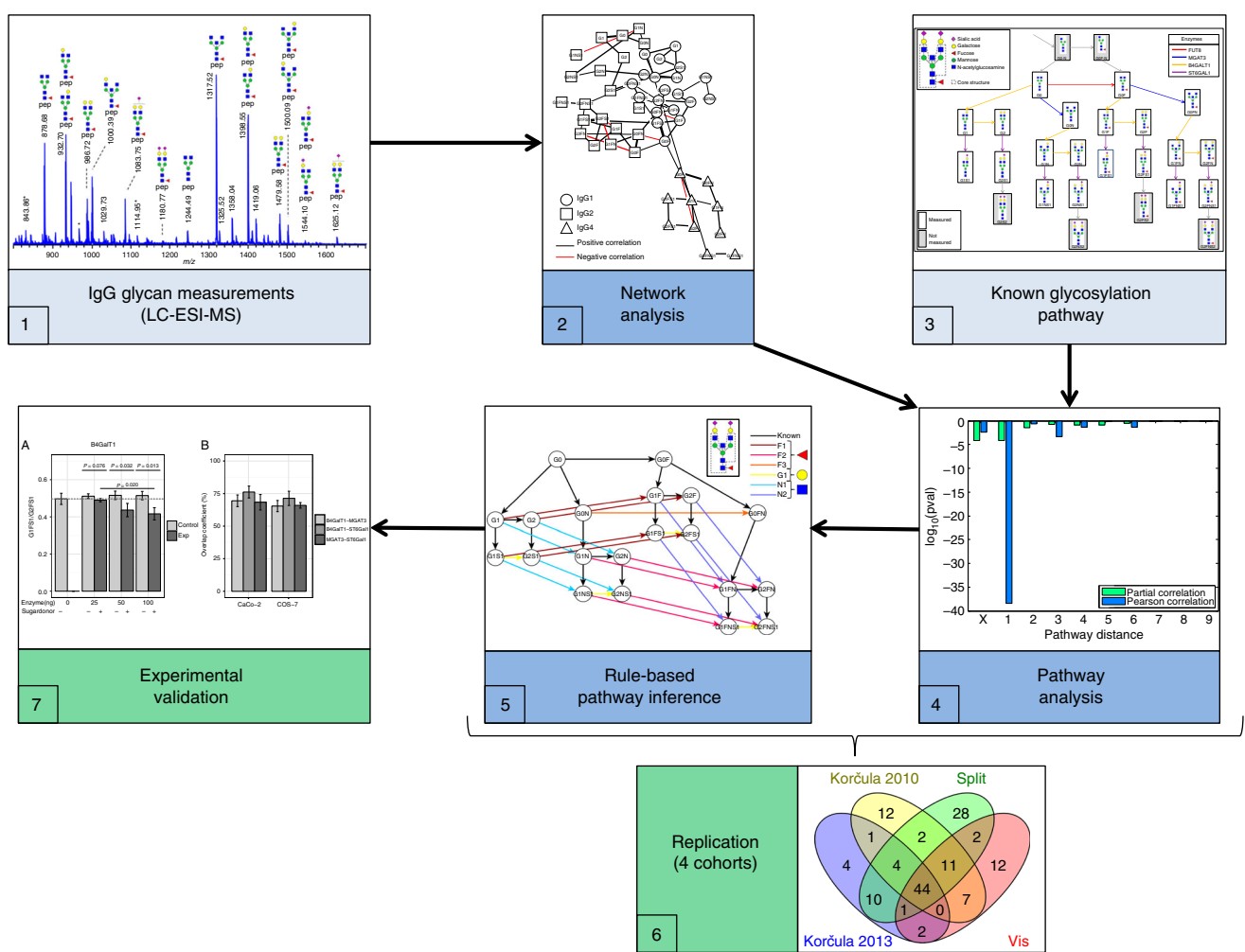

**Fig. 1** Analytical procedure. Starting from the IgG glycan abundances measured using LC-ESI-MS (1), we calculated a correlation-based network (2) and mapped it to the known IgG glycosylation pathway (3). We found that most edges in the network corresponded to single enzymatic steps in the pathway (4). Based on this finding, we inferred unknown enzymatic reactions that were putatively involved in the synthesis of IgG glycans using a rule-based approach (5). We then replicated the findings using four cohorts (6) and performed different in vitro validation experiments to confirm the predicted reactions (7)

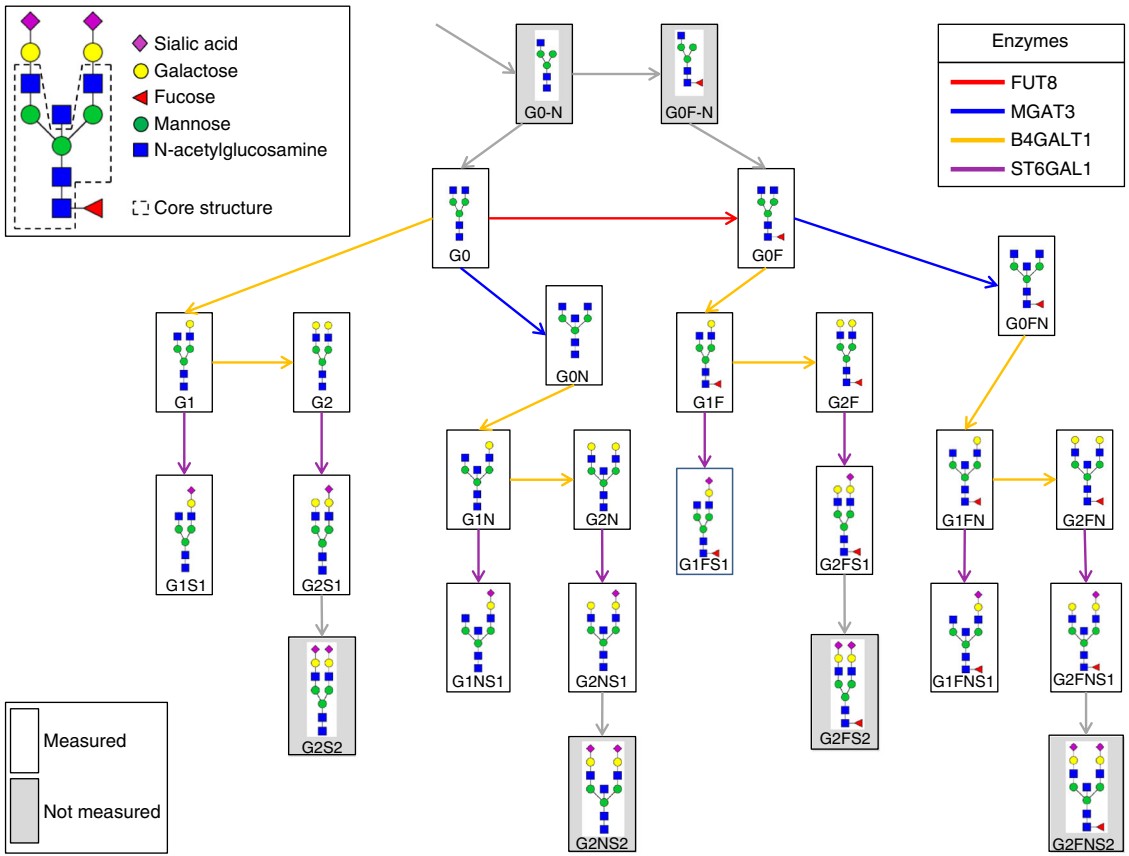

**Fig. 2** IgG glycan structures and the known glycosylation pathway. IgG glycans are biantennary complex-type structures: all measured glycoforms have a common core structure (the G0 structure, top left) to which additional sugars can be attached. The glycan names describe how many galactoses (G0/G1/G2) are present, whether there is a core fucose (F) or a bisecting N-acetylglucosamine (also referred to as GlcNAc, N), and whether the structure includes one or two sialic acids (S1/S2). The known glycosylation pathway as described in the review of Taniguchi[58] is shown, with boxes representing glycan structures and arrows representing single enzymatic reactions in the synthesis process. Each enzymatic reaction involves the addition of a single sugar unit to the glycan structure. The 20 glycoforms measured in this study are produced by four enzymes (FUT8, B4GalT1, MGAT3, and ST6Gal1). The gray boxes represent IgG glycoforms that were not measured in the present study. For a list of the primary literature describing the in vitro experiments which the pathway is based on, see Supplementary Fig. 1

represent their pairwise correlations, corrected for the confounding effects of all other glycans and clinical covariates. Previous studies using serum metabolomics data have shown that highly correlated pairs in GGMs represent enzymatic reactions[13,14]. This is the first study to apply GGMs to large-scale IgG glycomics data from four independent populations. We find that significant partial correlations predominantly occur between glycan structures that are one enzymatic step apart in the known IgG glycosylation pathway shown in Fig. 2, demonstrating that network statistics on quantitative glycoprotein measurements allow us to detect true enzymatic reaction steps in the glycosylation pathway.

Based on this result, we expect edges in the GGM that did not appear in the known pathway to represent true but hitherto unknown enzymatic steps, i.e., unknown substrate specificities of the enzymes in the pathway. To investigate this hypothesis, we develop a rule-based inference approach to test alternative pathway models. This shows that additional reactions are supported by the data for all four cohorts. More in detail, we predict that bisection of fucosylated, galactosylated glycans, as well as galactosylation of monosialylated glycans occur during IgG glycan synthesis. As direct experimental validation is considered unfeasible for the reasons outlined above, we validate our findings with two different approaches. First, we use a genome-wide association study (GWAS) in a fifth cohort. It has previously been

shown that the substrate–product ratios of metabolites are associated with their enzymes in GWAS[15,16]. Therefore, we consider the ratios of substrate–product pairs of the predicted reactions as quantitative traits, with which we can confirm several of our predicted reactions across the IgG subclasses. Second, we perform three sets of in vitro experiments to confirm the predicted enzymes substrate specificities, as well as their colocalization inside the Golgi apparatus. Our results show that at least one of the inferred reactions occurs in vitro, that one rejected rejection does not occur, and that the glycosyltransferases involved in the predicted reactions are colocalized in the Golgi stacks of two different cell lines.

## Results

**IgG glycomics correlation networks**. We measured the abundances of plasma IgG Fc N-glycopeptides in four Croatian cohorts—two from the island of Korčula (one sampled in 2010 and one in 2013), one from the island of Vis, and one from the city of Split (Table 1)—using liquid chromatography coupled with electrospray mass spectrometry (LC-ESI-MS). For each individual, we detected the abundances of 50 glycoforms—20 for IgG1, 20 for IgG2 and IgG3 combined (hereafter referred to as IgG2), and 10 for IgG4 (see Methods). In total, 20 different glycan structures were measured. Non-fucosylated glycans were not quantified for IgG4 due to their low abundances.

**Table 1 Characteristics of the four cohorts analyzed**

| | Korčula 2013 discovery | | | Korčula 2010 replication | | | Split replication | | | Vis replication | | | KORA validation | | |
|---|---|---|---|---|---|---|---|---|---|---|---|---|---|---|---|
| Number of measured glycans | 50 | | | 50 | | | 50 | | | 50 | | | 50 | | |
| IgG1    IgG2    IgG4 | 20 | 20 | 10 | 20 | 20 | 10 | 20 | 20 | 10 | 20 | 20 | 10 | 20 | 20 | 10 |
| Total number of samples | 695 | | | 951 | | | 994 | | | 780 | | | 1823 | | |
| Males    Females | 277 | 418 | | 339 | 612 | | 390 | 604 | | 326 | 454 | | 888 | 935 | |
| Samples with no missing values | 669 | | | 849 | | | 980 | | | 729 | | | 1641 | | |
| Unrelated samples | 669 | | | 504 | | | 980 | | | 395 | | | 1641 | | |
| Males    Females | 271 | 398 | | 156 | 348 | | 386 | 594 | | 152 | 243 | | 793 | 848 | |
| Age range | 18–88 | | | 18–90 | | | 18–85 | | | 18–91 | | | 32–81 | | |
| (median,IQR) | (52,24) | | | (56,18) | | | (52,21) | | | (54,24) | | | (61,14) | | |
| (mean,std) | (53,16) | | | (56,14) | | | (50,14) | | | (55,15) | | | (61,9) | | |

The Korčula 2013 cohort was selected for use in the discovery analysis. The results for all other cohorts are discussed in the replication section below. Glycan measurements were obtained for 695 individuals (277 men and 418 women) in the discovery cohort. Following preprocessing, which included data normalization and the removal of missing values and related individuals from the data set (see Methods), 669 samples (271 men and 398 women) remained, with an age range of 18–88 years.

We used both regular Pearson correlation and partial correlation analysis to make comparisons. The partial correlation analysis tested the conditional dependency between two variables when accounting for the confounding effects of all other glycans, as well as age and gender. In total, 905 Pearson correlation coefficients were significantly different from zero following multiple testing correction (false discovery rate (FDR) = 0.01). This significance level corresponded to an absolute correlation cutoff of 0.105, with coefficients approximately symmetrically distributed around zero (Supplementary Fig. 2A). Partial correlation coefficients are, by nature, much lower in absolute value than Pearson coefficients, and so only 66 of the total 1275 coefficients were found to be significant, the majority of which were positive (Supplementary Fig. 2B).

Upon inspection of the correlation matrices, we observed a remarkably similar structure between the different IgG subclasses (Supplementary Fig. 2C, D)—that is, glycoforms that were strongly correlated in one subclass also tended to be strongly correlated in the other subclasses. Moreover, there were only a few significant correlation coefficients for cross-subclass glycan pairs (off-diagonal blocks of the matrix). This suggests that the regulation of IgG is highly conserved across subclasses. Interestingly, seven of the nine cross-subclass pairs involved glycans with the same structure. For a full list of the partial correlations, see Supplementary Data 1.

The Pearson and partial correlation matrices were represented as networks (i.e., weighted graphs), with the nodes representing glycans and the edges indicating statistically significant coefficients (Fig. 3a, b). Most of the network edges connected glycan pairs that differed by only a single monosaccharide residue. This directly reflects the underlying glycan synthesis pathway, whereby monosaccharides are added one at a time and in a given order to create the final glycoform (Fig. 2). Furthermore, unlike the GGM, which showed a strong modularity with respect to the IgG subclasses, the Pearson correlation network did not show a clear separation between the subclasses (Fig. 3c, d). This indicates that significant partial correlations were mostly found between glycans belonging to the same IgG subclass, with few significant correlations between glycans with different IgG isoforms. To investigate this observation quantitatively, we calculated a

subclass-based network modularity for all significantly positive edges based on the method by Newman[34], and as previously adapted by Krumsiek[13]. We used degree-preserving random edge rewiring as a null model to assess the statistical significance (see Methods). The computed modularity for the original network was $Q = 0.495$ with an empirical $P$-value of $<10^{-5}$, proving a high level of subclass-specific modularity.

**Overlap of GGM with known IgG glycosylation pathway**. We systematically investigated the relationship between the known IgG glycosylation pathway (Fig. 2) and the data-driven GGM (Fig. 3b). To do this, we defined the "pathway distance" between any pair of glycans as the minimum number of enzymatic steps separating the two structures—for example, two glycans that corresponded to the reactant and product of a single enzymatic reaction in the IgG glycosylation pathway had a pathway distance of 1, whereas the shortest path from G0 to G2S1 includes three enzymatic steps, giving them a pathway distance of 3. We could not interpret correlations between glycans with the same structure belonging to different IgG subclasses in terms of the enzymatic reactions because they are bound to different proteins, and so we labeled these "X" (Fig. 4). All other cross-subclass glycan pairs were ignored in our analysis.

Significant Pearson correlation coefficients were found for both short and longer pathway distances (Fig. 4a); however, there were far more significant partial correlation coefficients at a pathway distance of 1 (Fig. 4b) than at any other pathway distance, demonstrating that significant partial correlations tend to occur between glycans that are directly connected in the pathway. To assess whether significant partial correlations occurred more often at a given pathway distance than expected by chance, we performed a Fisher's exact test. The results of the test were highly significant ($P = 3.41 \times 10^{-39}$; Fig. 4c, d), proving that there is a strong relationship between the data-driven GGM and the known IgG glycosylation pathway.

**Rule-based prediction of new enzymatic reactions**. Above, we demonstrated that significant partial correlation coefficients represent pairs of glycans that are directly linked in the known IgG glycosylation pathway. Interestingly, however, there were also 22 significant partial correlations for pathway distances greater than 1 (and not contained in the "X" group), as indicated by the black oval in Fig. 4b. Therefore, given the strong evidence for a relationship between the GGM and the IgG glycosylation pathway, we hypothesized that these correlations represented true but yet unknown pathway reactions. In principle, all glycans that differ in structure by a single monosaccharide could be connected

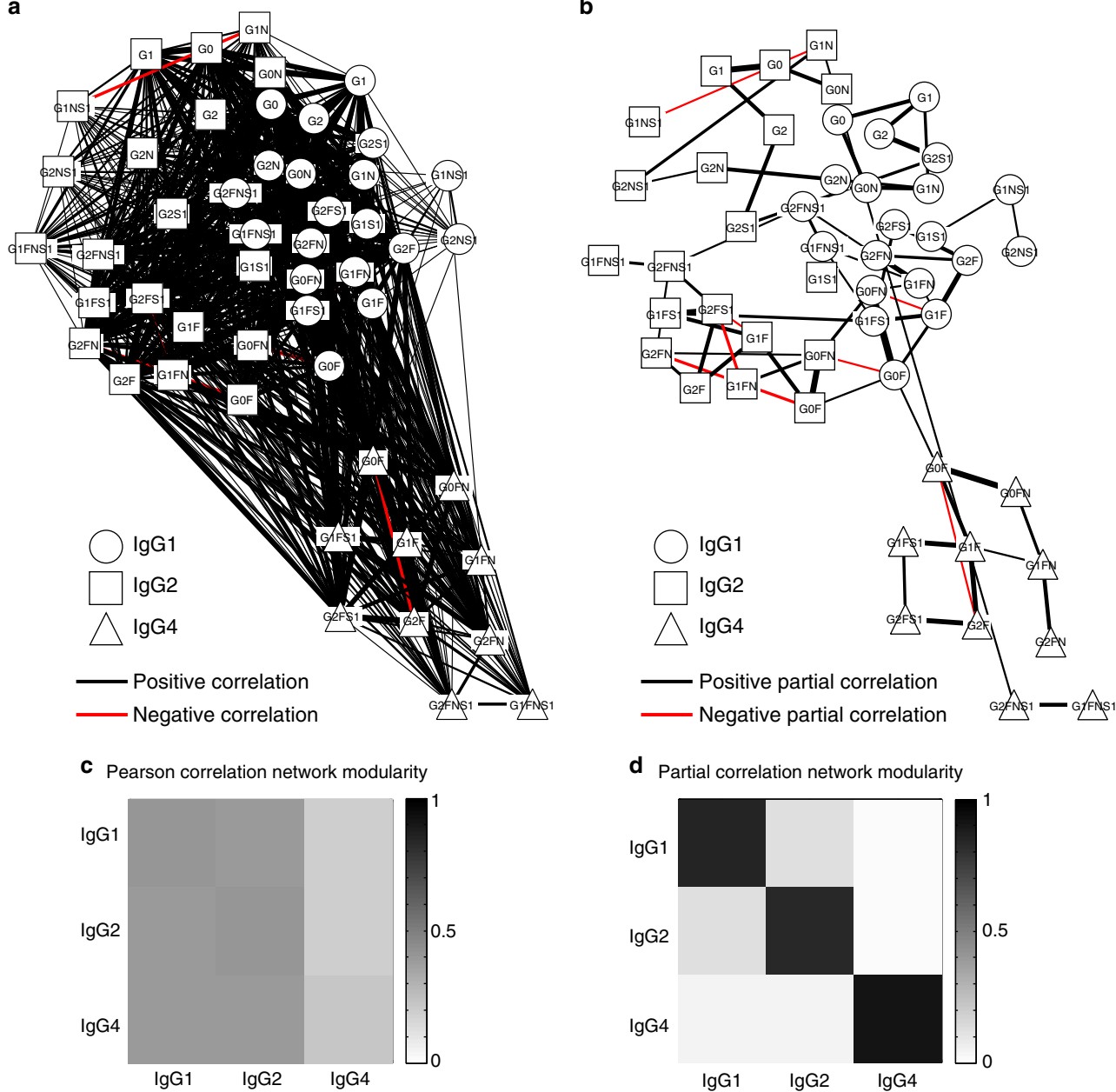

**Fig. 3** Network representation and modularity. **a**, **b** Pearson and partial correlation matrices, respectively, visualized as networks, where the nodes represent different glycoforms, and the edges indicate significant positive (black) and negative (red) correlations. Different node shapes correspond to different IgG subclasses, while the thickness of each edge corresponds to the magnitude of the respective correlation. **c**, **d** Pearson and partial correlation modularity, respectively, between IgG subclasses, measured as the relative out-degree from each subclass (row) to each other subclass (column). A Pearson correlation modularity analysis showed that all subclasses were highly interconnected. By contrast, the GGM showed high subclass modularity, indicating that associations between glycans mostly occurred within each IgG subclass. Furthermore, while the first two IgG subclasses were slightly interconnected, the Ig4 subclass was mostly isolated in the network

by a reaction performed by one of the four enzymes involved in the glycosylation pathway shown in Fig. 2; and among these 22 significant correlations, 15 (68%) differed by only one sugar residue. Furthermore, if we discard the seven negative partial correlations, whose interpretation has been shown to be problematic[13], this increases to 88% (Supplementary Data 1). Thus, all 15 of these glycan pairs are candidates for direct enzymatic reactions.

To analyze this quantitatively, we tested whether these unexplained partial correlations could be attributed to missing steps in the known pathway. To do this, we first created a list of

all possible novel pathway reactions, i.e., all connections between glycan structures that only differed by a single sugar unit and that were not present in the known IgG glycosylation pathway. Since we followed an unbiased approach, this included reactions for which in vitro experiments showed evidence of inhibition, e.g., the addition of fucose to the G0N structure[17]. We then divided these initial reactions into sets of "rules" according to the features of the hypothetical substrate and the corresponding enzyme performing the reaction (Fig. 5a and Table 2)—i.e., we built the rules to account for previously undescribed substrate specificities for the four glycosyltransferases involved in IgG glycosylation.

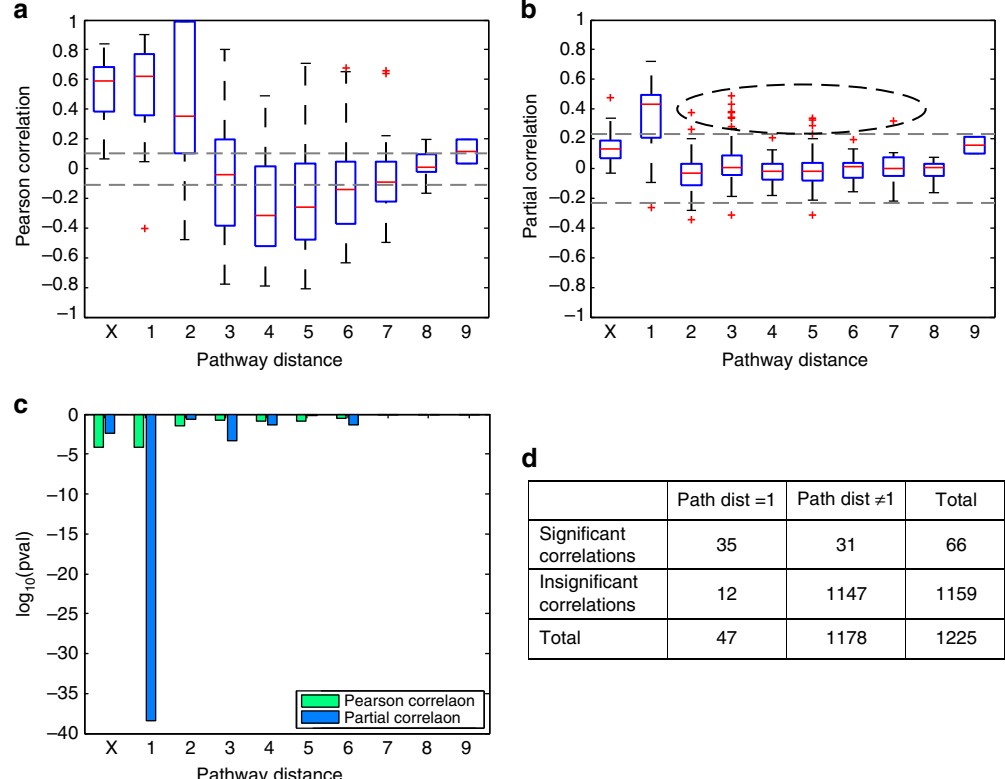

**Fig. 4** Systematic comparison of correlations and pathway distances. **a**, **b** Pearson and partial correlation coefficients, respectively, vs. pathway distance. Gray dashed lines represent the significance threshold (FDR = 0.01). On each box, the central mark indicates the median, and the whiskers indicate the 25th and 75th percentiles, respectively. The label "X" represents correlations between the same glycoforms across different IgG subclasses. In line with the network visualization, we observed significant Pearson correlation coefficients across all pathway distances, suggesting that Pearson correlations are non-specific with respect to the IgG glycosylation pathway. By contrast, significant partial correlation coefficients accumulated at a pathway distance of 1. The black dashed oval highlights significant partial correlations for pathway distances >1. **c** P-values for Fisher's exact tests for both Pearson and partial correlations at different pathway distances. There were significantly more significant partial correlations between glycans with a pathway distance of 1, demonstrating a close relationship between the IgG glycosylation pathway and the reconstructed GGM. The log10 P-values can be interpreted as a variance-normalized measure of the effect size. **d** Contingency table for the partial correlations at a pathway distance of 1. Entries represent the numbers of partial correlations satisfying the corresponding conditions

For example, the first rule (F1) describes the fucosylation of galactosylated, non-bisected glycans, as these reactions are not included in the known pathway. In this way, starting from 22 single potential new reactions, we defined six rules, as described in Fig. 5a and Table 2.

The rationale for our inference method and model selection technique was that the pathway model that contains the greatest proportion of true reactions should produce the lowest P-value with a Fisher's exact test, as seen in Fig. 4c. In total, we considered 63 pathway models that extended the known glycosylation pathway using all combinations of the six rules described above. To obtain a robust model fit, we performed bootstrapping with 10,000 resamplings and calculated 95% confidence intervals for each P-value distribution. We considered a pathway model to fit the data significantly better than the known pathway if it had a lower Fisher's test P-value and its 95% confidence interval did not overlap with that of the known pathway. Where several proposed pathway models were found to perform significantly better than the known pathway, we chose the simpler model, i.e., the one that included the fewest rules. Note that for this analysis we used P-values as variance-normalized measures of effect size for model comparison, rather than as the probability of an event occurring by chance. Figure 5b shows a comparison of the P-values for the known pathway, the known pathway extended with any one of the six defined rules, and all combinations that gave a significantly better P-value than the known pathway alone. A list with the results for all 64 (2^6) pathway models, including the known pathway for reference, can be found in Supplementary Data 2.

In the selected pathway model from this analysis, rules G1 and N2 were added to the known pathway (Fig. 5c), which resulted in the inclusion of eight new enzymatic steps in the IgG glycosylation pathway. By considering this selected model as the ground truth and reclassifying all partial correlations according to the pathway distances derived from this extended model, we found that most of the significant partial correlations that had longer distances in the original IgG glycosylation pathway (Fig. 4b) had a pathway distance of 1 in the modified IgG glycosylation pathway (Fig. 5d). Note that the pathway model that included all possible enzymatic reactions (model "F1F2F3G1N1N2" in Fig. 5b) did not yield the lowest P-value, indicating that the addition of more reactions than required to provide the optimal pathway model impaired the result.

**Replication in three additional cohorts**. We replicated these findings using IgG glycomics data measured on the same platform in three independent Croatian cohorts (Table 1). We again observed that most partial correlation coefficients between glycans were positive (Supplementary Fig. 3) and that the calculated

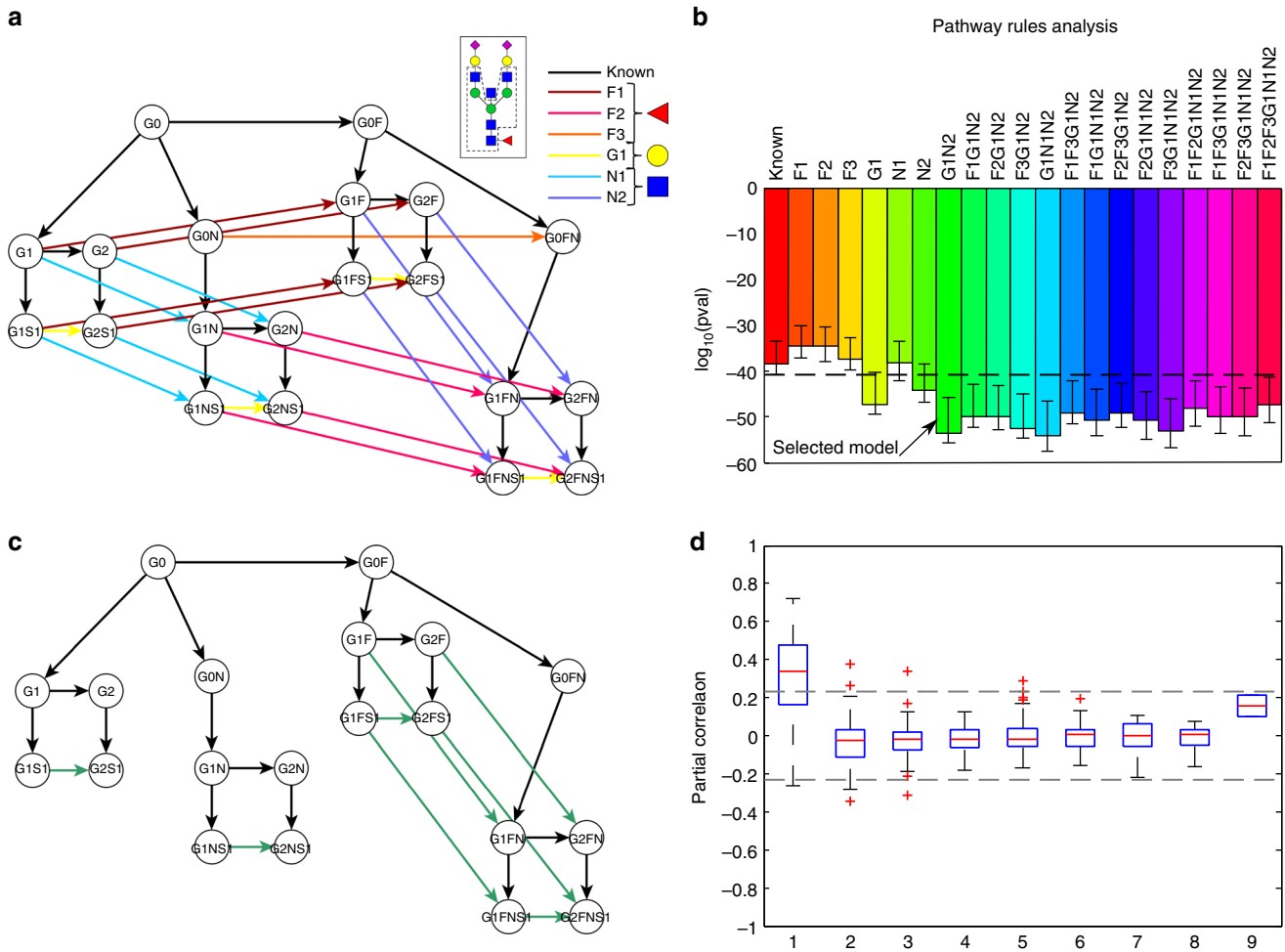

**Fig. 5** Rule-based approach for pathway inference. **a** Sketch of all single-monosaccharide additions in the IgG glycosylation pathway. The black network represents the known IgG glycosylation pathway, while arrows with the same color describe a rule. Shades of the same color represent reactions performed by the same enzyme. **b** Fisher's exact test results for the addition of different combinations of rules to the known pathway. The pathway model that most resembles the biological truth is expected to have the best overlap with the calculated GGM and hence yield a lower P-value. The black dashed line represents the lower end of the 95% confidence interval of the P-value for the known pathway obtained by bootstrapping. The simplest model that was significantly more accurate than the known pathway is indicated by a black arrow and includes rules G1 and N2. For a full list of results, see Supplementary Data 2. **c** Pathway model inferred by our approach. **d** Partial correlation coefficients for different pathway distances in the selected model. On each box, the central mark indicates the median, and the whiskers indicate the 25th and 75th percentiles, respectively. Most of the significant partial correlations that were a long distance apart in the known pathway are now at a distance of 1 (cf. Fig. 4b)

GGM displayed a highly modular structure with respect to the IgG subclasses (Supplementary Fig. 4). Moreover, pathway analysis showed that the edges represented single enzymatic steps in the IgG glycosylation pathway in all GGM networks (Supplementary Fig. 5). When inferring possible additional enzymatic steps, we again found that the addition of rules G1 and N2 to the known pathway gave a significantly better overlap with the GGM (Supplementary Fig. 6), providing further evidence that the enzymatic reactions included in these rules represent true steps in the IgG glycosylation pathway. The GGMs for all cohorts can be found in Supplementary Software.

To quantitatively evaluate the agreement between GGMs across the four cohorts, we generated a consensus network that represented the overlap between the networks (Fig. 6a). We considered an edge to be "replicated" if it was significant in all four cohorts. This showed that 44 of the 140 significant correlations were replicated across all four cohorts (Fig. 6b). To investigate how these edges related to the IgG glycosylation pathway, we again performed a Fisher's exact test. We only considered partial correlations that were found to be significant in

at least one cohort, and built a contingency table that classified these according to their replication status and pathway distance (Fig. 6c). The highly significant result of this test ($P = 7.73 \times 10^{-12}$) indicates that the replicated edges tend to represent true pathway reactions even more strongly than the non-replicated edges do, demonstrating that partial correlations corresponding to single enzymatic reactions in the IgG glycosylation pathway were robustly identified in all cohorts.

**GWAS evidence for predicted reactions.** We applied a GWAS-based approach on an independent cohort to provide evidence-based validation, assuming that significant associations between glycan ratios and single-nucleotide polymorphisms (SNPs) in the IgG glycosyltransferase genes indicate that the underlying reactions truly exist. This rationale is based on previous studies on blood metabolomics data, in which ratios of two metabolites were frequently found to be associated with genetic variation in the gene region of their catalyzing enzymes (see, e.g., Gieger[15]; Suhre[16]; Shin[14]). To quantify the increase of association strength

**Table 2 Rules for pathway inference**

| Substrate | Enzyme | | Product | Rule name |
|---|---|---|---|---|
| Galactosylated, non-bisected, non-fucosylated | FUT8 | ◀ | Galactosylated, non-bisected, fucosylated | F1 |
| Galactosylated, bisected, non-fucosylated | FUT8 | ◀ | Galactosylated, bisected, fucosylated | F2 |
| Non-galactosylated, bisected, non-fucosylated | FUT8 | ◀ | Non-galactosylated, bisected, fucosylated | F3 |
| Monogalactosylated, monosialylated | B4GalT1 | ⬤ | Digalactosylated, monosialylated | G1 |
| Galactosylated, non-bisected, non-fucosylated | MGAT3 | ■ | Galactosylated, bisected, non-fucosylated | N1 |
| Galactosylated, non-bisected, fucosylated | MGAT3 | ■ | Galactosylated, bisected, fucosylated | N2 |

of the ratio with respect to the single glycans, *p-gains* as defined in ref. [18] were used. Only significantly associated ratios with a sufficient p-gain (see Methods) were considered to confirm a given enzymatic reaction.

For this analysis, we used glycomics data from the German population study KORA F4[19]. Plasma IgG Fc *N*-glycopeptide measurements were obtained using the same LC-ESI-MS platform as for the discovery and replication cohorts described above, and included the same 50 measured glycoforms. Linear associations with genetic variants were calculated using the logarithm of all glycan product–substrate ratios defined in Fig. 5a (see Methods).

We considered SNPs in the four glycosyltransferase genes involved in IgG glycosylation (*ST6GAL1*, *B4GALT1*, *FUT8*, and *MGAT3*, see Supplementary Table 1). As a positive control, we first verified that the glycan product–substrate ratios in the known pathway were significantly associated with loci in the regions coding for the enzymes that are catalyzing the reactions. We found that 12 out of 47 ratios were genome-wide significant, while another five met a suggestive *P*-value of $10^{-7}$ (Fig. 7, thick black lines). Interestingly, we also found one ratio (G2/G1 in IgG2) that was associated with genetic variants in the region of the enzyme FUT8, which is responsible for the addition of core fucose (Fig. 7, arrow with asterisks). This was unexpected as neither of the structures in the ratio are fucosylated.

For our 22 predicted reactions, we found three significant and three suggestive hits (Fig. 7, thick green lines). Importantly, these significantly associated ratios tended to be the same across the three IgG subclasses and were equally distributed across the predicted rules. We found three confirmations for the rule G1 and three for the rule N2. By contrast, five significant associations and one suggestive hit were observed among the 26 ratios that were not predicted by our approach, but these did not replicate across subclasses (Fig. 7, thick gray lines). In particular, three out of these six non-predicted reactions originated from rule F1 exclusively in IgG1 and could not be replicated in IgG2, while the other three hits spread across three different rules.

Overall, we found at least one genome-wide significant association for all of the considered genes, providing evidence that we could indeed investigate all four glycosyltransferase enzymes involved in IgG glycosylation. The complete GWAS results can be found in Supplementary Data 3, while the regional association plots for visual inspection are provided in Supplementary Fig. 7.

**Experimental validation by enzymatic assays**. To address different aspects of our predictions, we performed three sets of in vitro experiments: two enzymatic assays and one colocalization experiment.

In a first experiment, we aimed to verify whether GalT1 and MGAT3 exhibited the predicted, previously unknown substrate specificities. To this end, we compared ultra performance liquid chromatography (UPLC) spectra of pooled IgG glycans before and after exposure to the two enzymes (see Methods). We considered seven different experimental conditions, covering various combinations of enzyme concentrations as well as negative controls (lacking sugar donors) that are not expected to show any reaction (Fig. 8a). As expected, GalT1 efficiently galactosylated a number of glycans in the IgG glycome (see Supplementary Data 4). To investigate our inferred reactions, we focused on the ratio of the substrate G1FS1 and product G2FS1. With increasing concentrations of added GalT1 enzyme (25, 50, and 100 ng), this ratio drops significantly compared to the respective negative controls (Fig. 8a), directly confirming one of the predicted reactions in rule G1 in a concentration-dependent manner. When performing the analogous experiment for MGAT3, however, we were not able to see addition of the bisecting GlcNAc to any of the glycans (not even reactions in the known IgG glycosylation pathway) (Supplementary Data 4). This might indicate that the fluorescent label attached to the IgG glycans interferes with the enzymatic reaction. For this reason, we were not able to experimentally prove or disprove any enzymatic reaction in rule N2.

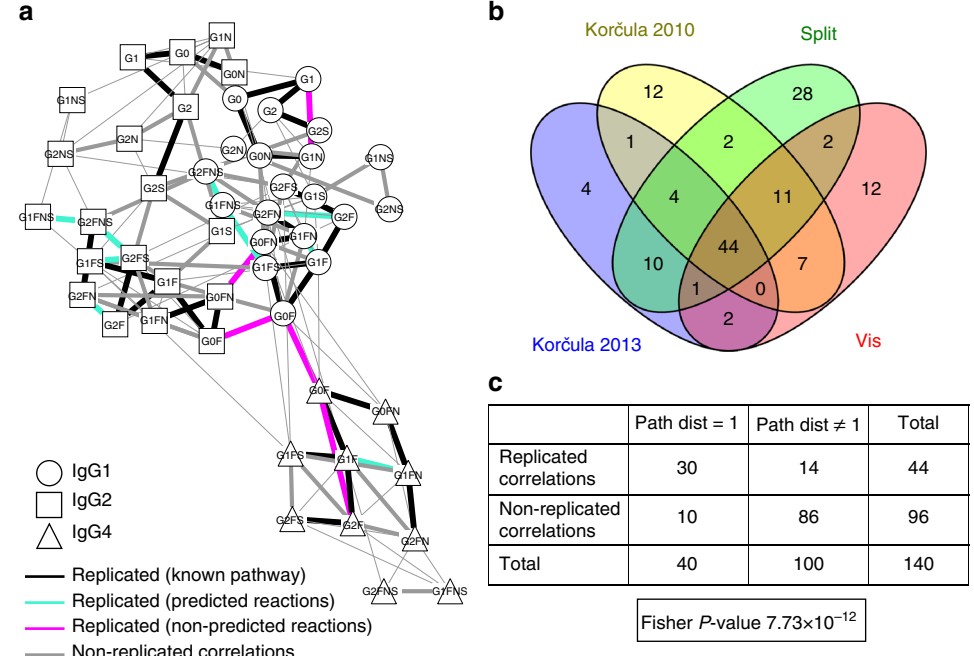

**Fig. 6** Replication. **a** Consensus network. Black edges represent replicated partial correlations that correspond to direct enzymatic steps in the known IgG glycosylation pathway, green edges represent replicated edges matching the reactions predicted by our approach, and red edges represent replicated correlations corresponding to reactions that were not predicted to take part in IgG glycosylation. Replicated edges were defined as partial correlations that were significant in all four cohorts. Gray edges represent partial correlations that were significant in at least one cohort but not in all four. Note that three of the five replicated but non-predicted edges linked the same glycan structure in different IgG subclasses, which we did not consider in our inference approach. Thus, there are only two edges that are truly non-predicted. **b** Venn diagram of the significant partial correlations in the four cohorts. In total, 44 edges were shared among all four cohorts. **c** Contingency table for the partial correlation coefficients that were found to be significant in at least one of the four considered cohorts. The classification variables in this case are replication status and pathway distance. Here, we considered edges that were significant in at least one of the four cohorts, and we considered an edge to be replicated if it occurred in all four cohorts. The resulting *P*-value was very low, indicating that replicated edges are more likely to represent enzymatic reactions than non-replicated edges

In a second experiment, we focused on a reaction from an excluded rule, namely the addition of bisecting GlcNAc to galactosylated non-fucosylated glycans (rule N1). To this end, we performed the enzymatic reaction with a pure G2 synthetic glycopeptide (see Supplemental Methods). For positive control, we also considered a G0 glycopeptide, a known substrate for MGAT3. While the G0 structure was completely converted to product within 3 h (Supplementary Fig. 8A), there was no measurable addition of bisecting GlcNAc to the G2 structure even after 48 h (Supplementary Fig. 8B). The latter result was further confirmed using a fluorescently labeled G2 glycan structure, which also did not show any conversion to G2N (Supplementary Fig. 8C). These experiments are thus supportive of our prediction that rule N1 is not enzymatically feasible.

**Enzyme colocalization experiments in cell lines.** In vitro evidence for enzymatic reactions does not necessarily translate to in vivo conditions. A general consensus in the field has been that Golgi glycosyltransferases mainly localize in the stack of cisternae according to their expected order of functioning[20,21]. In particular, this is expected to prohibit the bisection of galactosylated glycans due to the different localization of the enzymes. In contrast, our predictions suggest that the addition of bisecting GlcNAc can occur also on galactosylated, fucosylated glycans (rule N2).

To address this aspect, we performed colocalization experiments of the enzymes involved in our predicted reactions, namely B4GalT1, MGAT3, and ST6Gal1, in kidney COS-7 cells and

CaCo-2 colorectal cancer cells. Evidence of such a colocalization between the three glycosyltransferases would indicate that our predictions are, in fact, not impossible. Localization of the enzymes in the Golgi stacks of cisternae (cis-, -medial, -trans) was assessed using confocal microscopy and Z-stack imaging with Venus-tagged or Cherry-tagged enzyme constructs expressed at modest levels both in COS-7 and CaCo-2 cells. The latter have the advantage of having Golgi stacks dispersed throughout the cytoplasm, facilitating colocalization analyses at the level of individual Golgi stacks, and thereby aiding interpretation of the imaging data. In addition, cells were stained with anti-GM130 cis-Golgi marker antibody.

The overlap between the enzymes and GM130 was on average 62%/73% (MGAT3), 55%/67% (B4GalT1), and 51%/67% (ST6Gal1) in CaCo-2/COS-7 cells, respectively (Supplementary Fig. 9). This means that, to different degrees, all three enzymes can be found in the cis-part of the Golgi. Higher overlap percentages were detected in COS-7 cells due to the more compact Golgi architecture in the cells. Comparing the overall localization of the three enzymes, we observed prominent colocalization. The overlaps were quantified as 69%/65% (B4GalT1-MGAT3), 76%/71% (B4GalT1-ST6Gal1), and 68%/66% (MGAT3-ST6Gal1) in CaCo-2/COS-7 cells, respectively (Fig. 8b, c). These data indicate that, unexpectedly, a substantial proportion of all these three enzymes are present in the same Golgi compartments, indicating that our newly proposed reactions described by rules G1 and N2 are compatible with enzyme localization inside the Golgi stacks of cisternae.

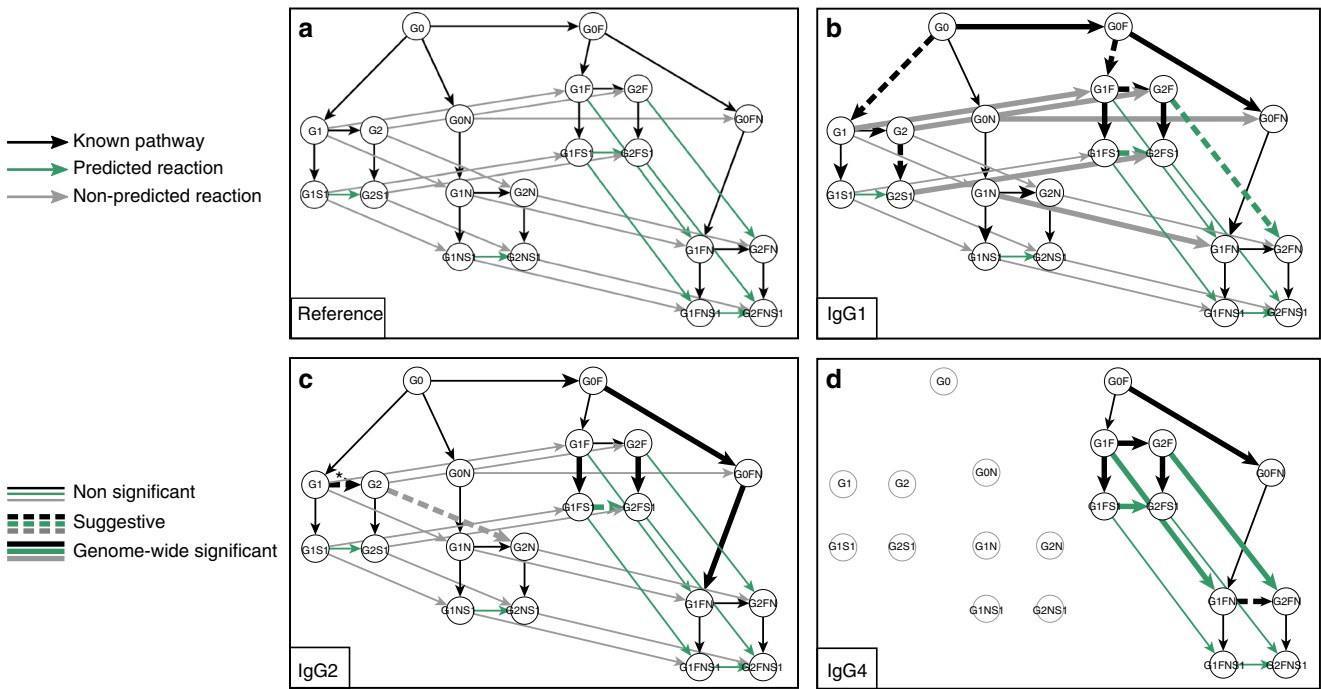

**Fig. 7** GWAS results for glycan ratios. **a** Reference pathway for interpreting the GWAS results. Black lines represent reactions in the known IgG glycosylation pathway, green lines represent reactions associated with the predicted rules, and gray lines represent possible reactions that were not selected by our approach. **b–d** GWAS results for IgG1, IgG2, and IgG4, respectively. Solid thick arrows represent ratios that were significantly associated with SNPs in the regions coding for an IgG glycosylation enzyme ($P < 5.26 \times 10^{-10}$ and p-gain > 10). Dashed arrows represent suggestive associations ($5.26 \times 10^{-10} \leq P \leq 10^{-7}$ and p-gain > 10). Gray nodes in the IgG4 plot represent glycoforms that were not measured. The asterisk indicates that the ratio was unexpectedly associated with SNPs in the FUT8 gene region

## Discussion

In this study, we demonstrated for the first time that GGMs can be used to reconstruct single enzymatic reaction steps in the glycan synthesis pathway using IgG Fc glycan measurements from human plasma. We also found that additional glycosylation reactions can be inferred from the calculated network, with the pathway rules G1 and N2 (Table 2) likely representing real biochemical steps in the IgG glycosylation pathway.

Rule G1 represents the galactosylation of sialylated glycans. The current standard glycosylation pathway is based on immunohistochemical studies that were performed over 30 years ago, which suggested different subcellular localization of galactosyltransferases and sialyltransferases in the Golgi apparatus[22,23]. Galactosylation is a pre-requisite for sialylation, and so the hypothesis of physically separated enzymes implied that galactosylation could only occur prior to sialylation. However, it has recently been shown that these two enzymes are colocalized in COS7 cells and are likely to act as a complex[24]. Our results directly support this hypothesis, as rule G1 represents sialylation of IgG prior to further galactosylation.

Rule N2 suggests that fucosylated, galactosylated glycans can be modified by adding a bisecting GlcNAc through MGAT3. Again, the standard glycosylation pathway assumes differential localization of the B4GalT1 and MGAT3 enzymes, and thus that the addition of bisecting GlcNAc could only occur prior to galactosylation. Previous studies have moreover indicated that overexpression of the B4GalT1 enzyme decreases the amount of bisecting GlcNAc (Fukuta et al. 2000), suggesting that the two enzymes might mutually inhibit each other's activity by competing for the same substrate. In contrast, our results suggest that the two enzymes may be colocalized and that galactosylated glycans could be direct substrates for MGAT3, hinting at a different regulation of the enzymatic activity than previously described.

As a limitation, it is to be noted that partial correlations calculated from glycomics data might not represent true biological processes in all cases. For example, we observed cross IgG-subclass correlations of the same glycan structures, which might be attributed to overall sugar or glycan abundances, rather than single enzymatic steps. Vice versa, not all glycan pairs that share a biochemical reaction will necessarily show correlation in the data. Reasons for this could be too low concentrations of glycans or high turnover rates of the IgG antibodies in blood. However, we used the correlation-based methodology to generate novel pathway hypothesis for experimental testing, which does not require a perfect reconstruction of the pathway.

Our findings were replicated across the analyzed cohorts, suggesting that the mechanisms that regulate IgG glycosylation are conserved across different Croatian populations. To validate our hypothesis for new enzymatic reactions in the IgG glycan synthesis pathway, we performed GWAS on glycan product–substrate ratios. Previous GWAS analyses on total IgG glycans measured with UPLC in the Vis and Korčula 2010 cohorts revealed statistically significant associations between traits describing fucosylated, non-bisected glycans, and the MGAT3 gene[9]. Here, we used specific glycan product–substrate ratios as quantitative traits, allowing us to analyze individual reactions at an IgG subclass-specific level. Six ratios that corresponded to our predicted reactions were found to be significantly associated with SNPs in the gene regions coding for the enzymes involved in the putative reactions (three for rule G1 and three for rule N2, see Supplementary Data 3), further supporting our hypothesis of these novel pathway steps occurring in vivo.

The GWAS evidence stems from an in vivo system; however, it is an indirect association and does not provide proof for the predicted reactions. Therefore, we performed in vitro enzyme assays probing specific reactions from the inferred pathway model. We found evidence that the addition of galactose to

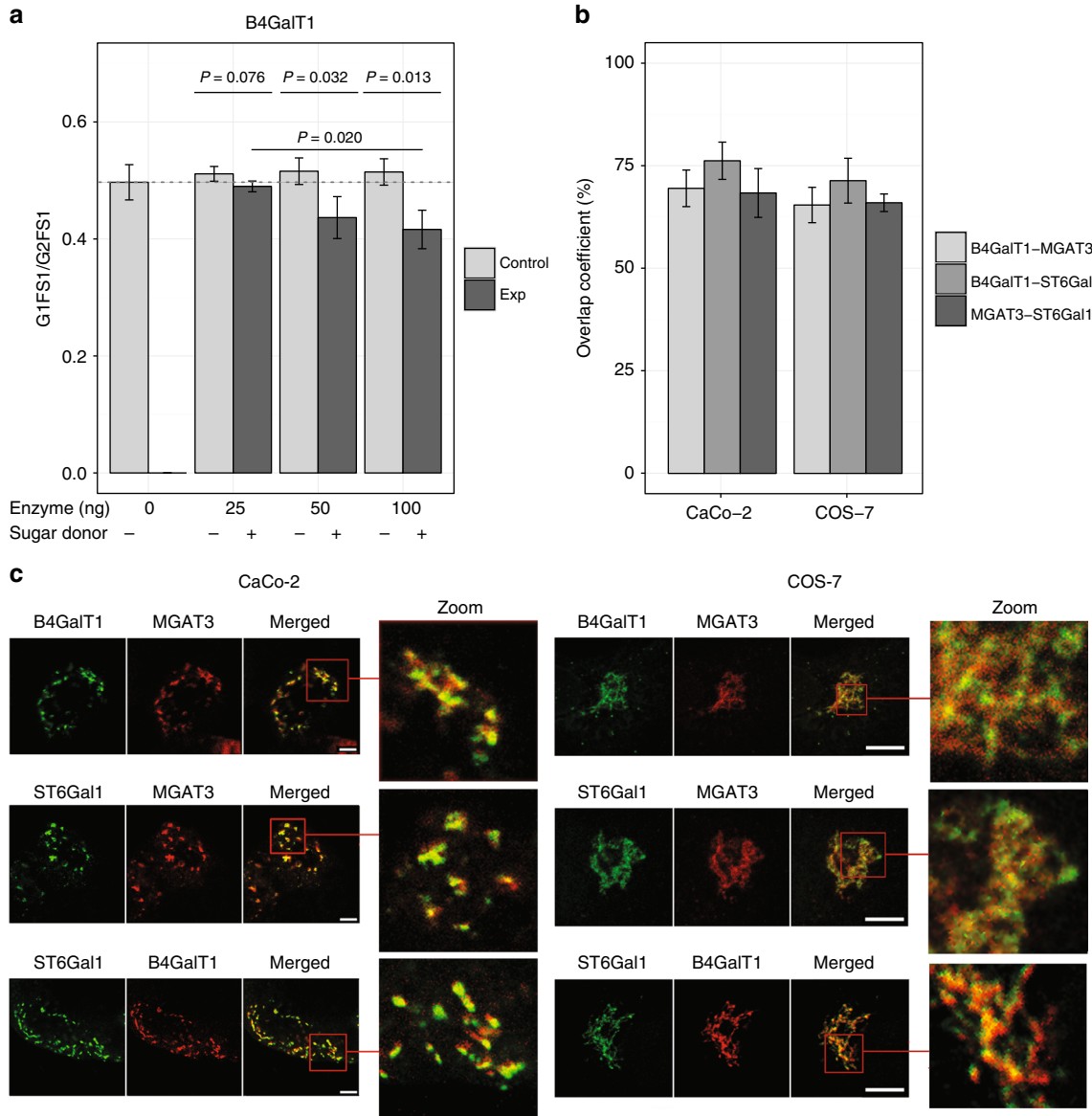

**Fig. 8** Experimental validation results. **a** In vitro enzymatic assay. The figure illustrates the ratio of G1FS1 over G2FS1 across different concentrations of the enzyme (B4GalT1), and in presence or absence of sugar donors. Bars represent the average value over triplicates, while error bars represent standard deviations. With increasing enzyme concentrations, the glycan ratio decreases significantly with respect to the corresponding negative control, confirming the occurrence of the predicted reaction. *P*-values were obtained from a two sample *t*-test. **b** Quantitative overlap between the localization of the three enzymes. The overall colocalization of each enzyme pair is expressed as an overlap coefficient percentage (mean % ± standard deviation). We observe substantial colocalization of all enzyme pairs in both cell lines. **c** Exemplary colocalization images of B4GalT1, ST6Gal-1, and MGAT3 in CaCo-2 (left) and COS-7 (right) cells, used for the overlap quantification in **b**. The individual figures represent a typical view from five different Golgi areas examined. In the images labeled as "Merged" and "Zoom", yellow areas represent enzymatic overlap. Due to the dispersed Golgi stacks throughout the cytoplasm in Caco-2 cells, the overlap can be observed clearly in separated cisternae, proving that localization of the glycosyltransferases is not limited to cis-Golgi, medial-Golgi, or trans-Golgi areas. Bar represents 5 μm

monosialylated glycans via B4GalT1 is indeed possible (rule G1). Confirming or disproving reactions in rule N2 was not possible due to experimental limitations. Moreover, we were able to show that a reaction from the rejected rule N1 did indeed not occur in the experiment. In addition to substrate specificity, current knowledge of the physical distribution of enzymes across the Golgi apparatus implied a directed order of the enzymes involved in the glycosylation process, thus preventing our predicted reactions from occurring in cells. In contrast to this, we found that the three enzymes involved in our predictions (B4GalT1, MGAT3, ST6Gal1) strongly colocalize across the Golgi in two different cell lines, suggesting that, in fact, the reactions are not unfeasible.

Taken together, while full in vivo validation of the new reactions is out of reach at this point, we found substantial evidence supporting our prediction in in vitro experiments.

Future studies could build on our findings in several ways. (1) The predicted rules could be investigated at a single-reaction level to determine whether all or only some of the enzymatic steps described in rules G1 and N2 are included in the IgG glycan synthesis pathway. (2) In addition, a single-reaction pathway inference approach could be used to explore the subclass-specific pathways suggested by some of our GWAS results. (3) The approach described in this paper could also be used to analyze other glycomics data sets, obtained from different platforms (e.g.,

UPLC fluorescence [FLR], matrix-assisted laser desorption ionization—time-of-flight—MS [MALDI-TOF-MS], or multiplexed capillary gel electrophoresis with laser-induced fluorescence detection [xCGE-LIF]; Huffman) to investigate whether the same reconstructed pathways are produced. (4) Measurement techniques for total plasma glycomes, including glycoforms with extremely heterogeneous structures (i.e., high mannose, hybrid, truncated, and complex glycans) from approximately 24 glycoproteins in blood, have recently become available[25]. Therefore, it would be of major interest to apply our methodology to these more complex data sets, to determine whether partial correlations can be used to reconstruct single enzymatic reactions even when dealing with a heterogeneous set of glycoproteins. (5) Replication of the results should also be verified in a non-Croatian cohort, as population-specific effects may have gone undetected in this analysis. (6) From a theoretical perspective, an analytical formulation of the likelihood function of the different pathway models based on information criteria such as the AIC (Akaike) or BIC (Bayesian) would lead to more rigorous model selection.

In conclusion, in this study we demonstrated for the first time that GGMs based on large IgG glycomics data sets contain strong footprints of biochemical reactions in the IgG glycosylation pathway. We proposed an inference algorithm based on the accordance of GGMs and the candidate pathways, to improve our understanding of the complex process of protein glycosylation. Novel reaction steps could be partially validated using GWAS data and in vitro experiments. In general, the finding that GGMs can be used to represent single steps in glycan synthesis indicates that it may be possible to compare the GGMs from healthy and sick individuals to detect alterations in enzymatic activity of the glycosyltransferases, shedding light on the molecular mechanisms that regulate IgG glycosylation.

## Methods

**Study populations**. In this study, plasma samples were obtained from five cohorts. Samples for the discovery and replication analyses came from the Croatian islands of Vis and Korčula, and were obtained from the "10001 Dalmatians" biobank[10], while samples for a second cohort from Korčula and a cohort from Split were collected separately a few years later (see Table 1 for details). For the purposes of the analysis described in this paper, we only considered unrelated individuals and samples with no missing values. Kinship coefficients[26] were estimated based on identity-by-state, which were computed using genotyped SNP data with the IBS function in the GenABEL package[27] for R. Unrelated individuals were obtained by selecting all pairs of individuals whose kinship coefficient was higher than 0.0312, which removed all individuals that were first-degree cousins or closer.

Validation was performed using the German study population KORA ("Kooperative Gesundheitsforschung in der Region Augsburg"; Wichmann[19]), which includes 3788 DNA samples with 18,185,628 SNPs (after QC) and 1887 glycomics samples. 1823 samples included both gene information and glycan concentrations and were considered in the analysis (Table 1).

All participants to the KORA study are residents of German nationality identified through the registration office and written informed consent was obtained from each participant. The study was approved by the local ethics committee (Bayerische Landesärztekammer). The Croatian cohorts received ethical approval of the ethics committee of the University of Split School of Medicine, as well as the South East Scotland Research. Written informed consent was obtained from each participant.

**Glycoproteomic measurements**. IgG was isolated from the plasma using affinity chromatography with 96-well protein G monolithic plates, as reported in the Supplementary Methods. IgG Fc glycopeptides were extracted through trypsin digestion and measured by LC-ESI-MS, which allows the separation of different IgG isoforms of glycans. In Caucasian populations, the tryptic Fc glycopeptides of IgG2 and IgG3 have identical peptide moieties[28,29] and so are not distinguishable using the profiling method. Furthermore, only 10 glycoforms of IgG4 were detectable due to the low abundance of this IgG subclass in human plasma. A detailed description of the experimental procedure can be found in the Supplementary Methods.

**Data preprocessing**. For each IgG subclass, the LC-ESI-MS raw ion counts were normalized using probabilistic quotient normalization, which was originally introduced for metabolomics measurements[30]. The reference sample was

calculated as the median value of each glycan abundance across all measured samples. For each sample, a vector of quotients was then obtained by dividing each glycan measure by the corresponding value in the reference sample. The median of these quotients was then used as the sample's dilution factor, and the original sample values were subsequently divided by that value. This procedure was repeated for each sample in the data set. To verify normal distributions, we compared QQ-plots against normal distributions for non-logarithmized as well as logarithmized glycan data (Supplementary Fig. 10). Since all distributions were closer to log-normality than to normality, we log-transformed the glycan concentrations prior to analysis.

**Correlation networks and modularity**. Correlation networks were computed using the preprocessed glycan abundances. Regular correlation networks are based on Pearson product–moment correlation coefficients, which represent the linear dependency between two variables. However, GGMs are based on partial correlation coefficients, which represent pairwise dependencies in multivariate normally distributed data when conditioned against all other variables. To obtain a reliable estimate for the partial correlation matrix, we used the shrinkage-based GeneNet algorithm[31]. All partial correlations were corrected for the confounding effects of age and gender. This is done by including the confounding variables in the GGM calculation, but not showing them as nodes in the final network.

Multiple hypothesis testing was corrected for by controlling the FDR at 0.01 using the Benjamini–Hochberg method[32]. All partial correlation coefficients between glycans that had non-significant Pearson correlation coefficients were omitted[33]. The final GGM is represented by all the significant partial correlation coefficients.

The network modularity algorithm was adapted from the widely used community detection clustering method of Newman[34], which optimizes a modularity Q to determine clusters. In this paper, we used the Q measure to assess the modularity of predefined clusters, given by the three IgG subclasses. To this end, subclass-based network modularity was calculated as the relative out-degree from each subclass to all other subclasses for all significantly positive edges. To assess the significance of the observed modularity, we performed graph randomization via edge rewiring[35,36]. In this process, two edges in the original data-driven network are randomly selected and the end nodes of each edge are swapped. The operation was repeated 10 times the number of edges to reach sufficient randomization. The entire randomization was repeated $10^5$ times to obtain a sufficient number of null model networks.

Computations were performed using Matlab R2014a and R 3.1.1.

**Pathway analysis**. Evidence on substrate specificities of the four enzymes involved in IgG glycosylation was based on in vitro experiments[17,37–48]. For more details, Supplementary Fig. 1 describes the primary literature for each enzymatic reaction in Fig. 2.

The pathway analysis and inference were performed using Fisher's exact tests, which evaluate whether two categorical variables are statistically independent[49,50], with low P-values indicating a lack of independence. For the purposes of this analysis, we tested whether significant partial correlation coefficients accumulated at given pathway distances.

**Bootstrapping**. To statistically compare alternative pathway models, we used bootstrapping to estimate 95% confidence intervals for the P-values calculated using Fisher's exact tests. To do this, we randomly resampled the original cohort to obtain a new data set with the same number of samples as the original. We then repeated the entire analysis pipeline, including the GGM calculation and pathway analysis, and obtained a new Fisher's P-value for each combination of pathway rules. Confidence intervals were based on 10,000 resampled data sets.

**Genome-wide association study**. Genotyping was performed using the Affymetrix GeneChip array 6.0 with prephasing by SHAPEIT v2 and imputation by IMPUTE v2.3.0, using 1000 Genomes (phase 1 integrated haplotypes CEU) as a reference panel. We limited our analysis to non-monomorphic SNPs that had a minor allele frequency >1%, a high genotyping quality (call rate >97%), and did not significantly deviate from the Hardy–Weinberg equilibrium (pHWE $\geq 5 \times 10^6$). Samples with mismatched phenotypic and genetic genders were excluded, leaving 1641 samples and 18,185,628 SNPs to be analyzed. All individuals were of European ancestry.

The glycan measurements were preprocessed using a similar pipeline as that for the Croatian data in the pathway analysis (see above). Samples from each IgG subclass were log-transformed and batch-corrected using the ComBat algorithm of the R package "sva" (R package version 3.14.0). The data were exponentiated to retrieve the original scale and then normalized using the probabilistic quotient algorithm[30]. Glycan ratios were calculated as the product–substrate ratios of all possible reactions in the IgG glycosylation pathway, as shown in Fig. 5a, and then log-transformed and regressed against age and sex. A rank-based inverse normal transformation was applied to the residuals.

For the purposes of this study, we only focused on SNPs located in the regions of the known glycosylation enzymes—ST6GAL1 (chr.3), B4GALT1 (chr.9), FUT8 (chr.14), and MGAT3 (chr.22)—and with a linkage disequilibrium (LD) and $R^2 \geq$

0.8 (Supplementary Table 1). Genomic positions were retrieved from the UCSC Genomic Browser (GRCh37/hg19)[51], while LD information was obtained using the software SNIPA[52].

GWAS was performed with snptest software v2.5.1[53] using an additive genetic model. We used an established GWAS significance threshold[54] corrected for the number of considered ratios, i.e., $\frac{5 \times 10^{-8}}{95} = 5.26 \times 10^{-10}$. For suggestive hits, we used a relaxed threshold of $10^{-7}$, as also suggested in ref. [54].

P-gains were introduced to describe the increase in strength of the association of a ratio compared to the corresponding single glycans. We assume that a significant P-value combined with a high p-gain indicates that the two glycans are functionally linked in a biochemical reaction involving the gene of the associating SNP. P-gains were defined as the ratio between the minimum of the association P-values of single glycans and the association P-value of the corresponding ratio[15,18]. The gain in the P-value of the ratio was considered to be significant if it was greater than or equal to 10, as this value indicates gains of one order of magnitude. This threshold was taken as suggestive also in previous studies, e.g., in Suhre[16] and Shin[14]. See Supplementary Data 3 for a full list of results.

**Validation of inferred reactions in vitro**. *Substrate preparation*: In-house pre-pared 2-aminobenzamide (2-AB)-labeled IgG glycans[55] were pooled and dried in a vacuum centrifuge to obtain the necessary amount for glycosyltransferase experiments. The preparation of G0-glycopeptide (G0-SGP) and G2-glycopeptide (G2-SGP) substrates from egg yolk sialylglycopeptide is reported in the Supplementary Methods. For details about the synthetic substrate preparation, see Supplementary Methods.

*Glycosyltransferase experiments*: Expression constructs were generated encoding the catalytic domains of human B4GALT1 (Uniprot P15291, residues 63–398) and human MGAT3 (Uniprot Q09327, residues 24–533) as an NH2-terminal fusion protein in the pGEn2 mammalian expression vector[56]. Enzymes were expressed in HEK293-F cells (FreeStyle™ 293-F cells, Thermo Fisher Scientific, Waltham, MA) and the secreted protein was purified by Ni2+-NTA (Qiagen, Valencia, CA) affinity chromatography by binding, and washing with a column buffer comprising 20 mM HEPES, 300 mM NaCl, 20 mM imidazole, pH 7.2 and eluted in the same column buffer containing 300 mM imidazole61. The sample was concentrated and loaded on a Superdex 75 gel filtration column (GE Healthcare) containing 20 mM HEPES, 200 mM NaCl, 60 mM imidazole, pH 7.2. The eluted GFP fusion proteins were concentrated to 1 mg/ml and used directly in enzymatic glycan modifications.

Enzyme activity assays were performed using the UDP-Glo assay (Promega). B4GALT1 assays were performed using 100 ng of enzyme, 1 mM acceptor (GlcNAc or G0-SGP), and 1 mM UDP-Gal. The MGAT3 assay was performed using 100 ng of enzyme, 1 mM acceptor (G0-SGP or G2-SGP), and 1 mM UDP-GlcNAc. The assay buffer for both enzyme assays contained 100 mM HEPES (pH 7.0), 2 mM MnCl2, 1 mg/ml BSA in a 10 μL reaction volume at 37 °C for 1 h. Reactions were stopped by mixing with an equal volume of UDP Detection Reagent (5 μL) and incubated for 60 min at room temperature. After incubation, luminescence measurements were performed using a GloMax Multi Detection System plate reader (Promega). Luminescence values were compared to a standard curve for quantification of UDP produced. Results were also confirmed using mass spectrometry of the enzymatic products using ESI-MS on a Shimadzu ESI-IT-TOF instrument.

*Analysis of IgG 2-AB glycans after glycosyltransferase experiments*: After the glycosyltransferase experiments, the reaction mixtures (10 μL) were purified using AcroPrep Advance 96-well filter plate, Omega 10K MWCO (Pall) using vacuum manifold (Pall) at maximum flow rate. The filter plate was washed twice with 100 μL of ultrapure water. After adding 50 μL of ultrapure water, each sample was resuspended, transferred to the filter plate and the eluate was collected in a clean PCR plate. This step was repeated twice. The filter plate was washed with additional 50 μL of ultrapure water and collected to the same PCR plate (total volume of 160 μL).

*Sample preparation for the HILIC-UPLC analysis*: Purified IgG 2-AB glycans modified with glycosyltransferases were analyzed by UPLC based on hydrophilic interactions (HILIC). Samples were prepared for the analysis by pipetting 20 μL of each sample and adding 80 μL of cold (4 °C) acetonitrile. Prepared samples were filtered through AcroPrep Advance 96-well filter plate, Omega 10K MWCO that was prewashed with 100 μL of ultrapure water using vacuum manifold at maximum flow rate. Samples (100 μL) were transferred to vials and 40 μL of each sample was injected to a column. IgG 2-AB glycans were analyzed on a Waters UPLC as described in the Supplementary Methods. Raw spectral data were then normalized using the probabilistic quotient method[30].

**Determination of enzyme colocalization in cell lines**. CaCo-2 (https://www.dsmz.de/catalogues/details/culture/ACC-169.html?tx_dsmzresources_pi5%5Breturn Pid%5D=192) and COS-7 (https://www.dsmz.de/catalogues/details/culture/ACC-60.html?tx_dsmzresources_pi5%5BreturnPid%5D=192) cells (n = 5 each) were grown in DMEM/10% FCS and transfected with the plasmids encoding the full-length B4GalT1, ST6Gal1, and MGAT3 tagged with either C-terminal mVenus or mCherry variant[57]. Transfections were carried out using Lipofectamine 3000 (Invitrogen, Thermo Fisher Scientific, Carlsbad, CA, USA), according to manufacturer's instructions, except that only 500 ng of the plasmids cDNAs were used for the transfections and the cells were fixed for immunofluorescence

microscopy already after 1 day. These procedures helped to keep the expression levels sufficiently low to avoid disturbing normal localization of the proteins in the Golgi. After fixation (4% paraformaldehyde/PBS, 20 min), blocking and permea-bilization (1% BSA, 0.1% saponin in PBS, pH 7.4) the cells were co-stained with the cis-Golgi marker antibody (monoclonal anti-GM130, Thermo Scientific, Cheshire, UK, 1:250 dilution, https://www.fishersci.com/shop/products/anti-gm130-clone-35-bd-2/bdb610822#?keyword=610822) together with Alexa Fluor 594-conjugated goat anti-mouse secondary antibodies (Molecular Probes, Eugene, OR, USA, 1:500 dilution, https://www.thermofisher.com/antibody/product/Goat-anti-Mouse-IgG-H-L-Cross-Adsorbed-Secondary-Antibody-Polyclonal/A-11020). After staining, cells were embedded with Immu-Mount (Thermo Scientific, Cheshire, UK) and stained specimens were examined and photographed using the Zeiss LSM710 confocal microscope with 100× oil immersion objective (n.a. = 1.3). 30–40 sections at 0.3 μm intervals were taken from the selected regions using the pinhole setting to 1. The signal intensities in each channel were quantified from each Z-stack image using the image quantification module of the Zen 2009 software. The same soft-ware package was used to calculate the overall colocalization of each enzyme pair or enzyme–Golgi marker pair. The values are expressed as overlap coefficient percentages (mean % ± standard deviation) obtained using pixel per pixel com-parison of each Z-stack image and by averaging the values for each set of Z-stack images. The presented values therefore illustrate the overall colocalization of the proteins throughout each Golgi structure examined.

**Data availability**. The informed consent given by KORA study participants does not cover data posting in public databases. However, data are available upon request from KORA-gen (http://epi.helmholtz-muenchen.de/kora-gen). Requests are submitted online and are subject to approval by the KORA board. Preprocessed glycan concentrations, as well as values corrected for age and gender, are available in the figshare repository with the identifier doi:10.6084/m9.figshare.5335861. Note that for confidentiality reasons, age and gender data cannot be shared publicly, and we therefore additionally provide the corrected glycan values. A GGM inferred on this corrected data matrix will be largely identical to the one presented in this paper.

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

## Acknowledgements

The CROATIA_Vis, CROATIA_Korčula, and CROATIA_Split studies were funded by grants from the Medical Research Council (UK), European Commission Framework 6 project EUROSPAN (Contract No. LSHG-CT-2006-018947), FP7 contract BBMRI-LPC (grant No. 313010), Croatian Science Foundation (grant 8875), and the Republic of Croatia Ministry of Science, Education and Sports (216-1080315-0302). We would like to acknowledge the staff of several institutions in Croatia that supported the field work, including but not limited to The University of Split and Zagreb Medical Schools, Institute for Anthropological Research in Zagreb, and the Croatian Institute for Public Health. This work was funded in part by grants from the German Federal Ministry of Education and Research (BMBF), by BMBF Grant No. 01ZX1313C (project e:Athero-MED), by the European Commission HighGlycan (contract #278535), MIMOmics (contract #305280), HTP-GlycoMet (contract #324400), IntegraLife (contract #315997), and CarTarDis (contract #602936) grants. N.B. was supported by the Russian Science Foundation (grant #14-50-00131). Additional support was from NIH grants P41GM103390 (to K.W.M.), P01GM107012 (G.J.B., PI). The KORA study was initiated and financed by the Helmholtz Zentrum München—German Research Center for Environmental Health, which is funded by the German Federal Ministry of Education and Research (BMBF) and by the Free State of Bavaria. We thank Erik van den Akker for useful comments and suggestions.

## Author contributions

E.B., G.L., F.J.T. and J.K. conceived and designed the project. M.P.-B., T.K., I.T.-A., G.R., J.S., L.K., I.U., M.H.J.S., M.W., I.R., O.P., C.H., H.G., K.S., A.P., T.M., C.G., G.L. contributed the data. E.B. performed the analysis on the Croatian cohorts and wrote the

primary manuscript. A.L. performed the GWAS analysis. I.T.-A., M.V. and G.L. prepared the pooled 2AB-labeled glycans and analyzed the UPLC spectra. N.B. and T.O. prepared the fluorescently labeled synthetic substrate. J.-Y.Y., L.L., G.-J.B. and K.W.M. performed the in vitro enzymatic reactions. A.H. and S.K. performed the localization experiments. All authors approved the final manuscript.

## Additional information

**Competing interests:** G.L. declares that he is a founder and owner of Genos, a private research organization that specializes in high-throughput glycomics and has several patents in the field. M.P.-B., I.T.-A., G.R., J.S., L.K. and M.V. are employees of Genos. The remaining authors declare no competing financial interests.

