## [Peer Review file · Nature Communications]

Reviewers' comments:

Reviewer #1 (Remarks to the Author):

The manuscript of is an awe-inspiring example of the the application of mathematical methods to "big" glycomics data. I should rightaway confess that I lack the background to fully assess the statistics and even less the genetic analyses.

However, I am surprised about the biosynthetic "rules" postulated by the authors as they all oppose the existing knowledge on the substrate specificity of the various glycosyltransferases. Given the fact that bisecting GlcNAc is used to produce non-fucosylated IgG on a large scale, it is surprising to learn about a pathway where galactosylated, bisected glycans are fucosylated (rule F2). In the discussion, the authors argue with possibly outdated models of enzyme location. In my view, this is not the point. The crucial point rather is the substrate specificity of the enzymes involved. The relevant glycosyltransferases have all been purified or expressed and amply characterized. Notably, the related literature does not seem to be cited. Certainly, no Harry Schachter paper is found in the literature list.

So, while the statistical apparatus applied to the data sets (whose quality I will not question) may be of finest nature, and although the fact that the accepted biosynthetic pathways in fact do emerge as "thick" edges in the model I am left with the uneasy feeling that this lemon has been squeezed out a bit too much.

If I am wrong, the authors should not have too much trouble to add some wet chemistry proving their new "rules" right. Then it would be a splendid work. Unless this is possible, it appears – excuse me – a bit as art for arts' sake.

Reviewer #2 (Remarks to the Author):

What follows is a review of the article entitled "Network inference from glycoproteomics data reveals new reactions in the IgG glycosylation pathway" by Benedetti et al and submitted to Nature Communications. This article describes a method that utilizes IgG1 Fc N-glycan composition to identify novel enzyme-catalyzed steps in glycan biosynthesis. The authors compared pathway analysis using a Pearson coefficient with a partial correlation coefficient and noted the partial correlation coefficient provided a superior result. The authors correctly identify a shortcoming in the in vitro descriptions of the Golgi N-glycan remodeling biosynthesis pathway: all potential N-glycan substrates and products of each enzyme are not known, furthermore, all potential IgG Fc-conjugated N-glycan substrates and products of each enzyme are not known. It is the latter the authors address and their data will likely inform the former as reactions catalyzed by the three glycosyltransferases identified (Fut8, B4GalT1 and MGat3) affect many N-glycoproteins. In general the manuscript was well written and clear. The points were clearly described and the conclusions are interesting. The statistical analysis is largely well explained, which this reviewer finds unusual and appreciates with respect to this manuscript.

Despite a fairly promising start with 22(?) new potential connections, the authors justifiably settle on only three novel biosynthetic routes. It should be mentioned that these are not novel biosynthetic pathways, just an identification of a new substrate for a particular enzyme (and not ones that are particularly surprising). Unfortunately this limits impact of the manuscript.

One other aspect limited the enthusiasm of this reviewer for the manuscript as presented. The authors note on line 203 "To the best of our knowledge, it is not feasible to prove the predicted reactions experimentally under in vivo conditions." That is probably correct. They neglect to mention that it is likely possible, with current technologies, to test the reactions in vitro. Such a proof of principle would represent a significant step towards validating the statistical methods. Furthermore, such a measurement, with adequate controls, would provide evidence an enzymologist would find interesting, most notably: are these unpredicted substrates utilized with reasonable enzyme kinetics? It does not appear from the manuscript that the authors are able to deconstruct relative flux through each of the pathways, which would likewise also improve the

manuscript and strengthen the impact. If so, these data should be discussed in a clear manner.
Minor notes:

Line 186: "We replicated this findings" should read "We replicated these findings"

Under what hypothetical circumstances could connections occur that do not represent an enzyme catalyzed reaction? Cross subclass connections are found and are physically meaningless, as noted. It would be valuable to discuss limitations of the technique more thoroughly.

Recent evidence from the Lau and Cobb groups independently indicates that galactosylation and sialylation of IgG Fc is spatially decoupled. It would be worthwhile to consider that argument in the paragraph starting on line 239 that notes a physical connection between the B4GalT1 and St6Gal1 enzymes in the Golgi.

Line 251: it is premature to suggest MGat3 and B4GalT1 are colocalized based on the network analysis data presented in the manuscript.

Unrelated humorous note: the "10001 Dalmatians biobank" is well named.

Reviewer #3 (Remarks to the Author):

Overview of the article:

The present study investigates data driven approaches for inferring IgG glycosylation pathway from large-scale mass spectrometry measurements. The authors argue that the current knowledge on glycosylation pathway is incomplete especially in in-vivo settings as in-vivo systems do not lend themselves for experimental manipulations. Subsequently, several computational strategies are presented in this regards by the authors. Pair-wise correlations, partial correlations, GGMs for identifying significant associations with a glycosylation pathway template using Fisher's exact tests, and correlating the findings to SNP data. The results are demonstrated on four croatian cohorts from different islands.

Specific Comments:

1. The authors should clear ambiguities related to the "benchmark pathway" since they compare their data driven approach to this benchmark. How did they arrive at the benchmarks? This question is especially pertinent since they critique existing in-vivo glycosylation pathways as incomplete and warranting the proposed data driven study.

2. Can glycosylation pathway vary across the four different croatian cohorts in the study? Is it important to have a whole genome sequencing across these cohorts to establish the association between potential sequence variations and its impact on the glycosylation?

3. Is the glycosylation pathway a directed cyclic or acyclic graph? i.e. can it have feedback.
Direct dependencies: Pair-wise correlations essentially models potential associations as undirected weighted graphs where weights may represent the magnitude of correlation. The authors have used pearson correlation in conjunction with FDR for modeling these. Fine.

Indirect Dependencies: On the other hand, dependencies between a pair of variables can be indirect and mediated through other variable(s) in which case one might model conditional dependencies using measures such as partial correlations as opposed to pairwise dependencies.

Questions and concerns:

1. If the objective is accommodate indirect dependencies then why are the results of Pearson correlation shown? Since dependencies modeled using partial correlation is a subset of those modeled using pearson correlation, it is well known that the corresponding graphs are less dense. So the figures on pearson correlation contribute no information and can be removed.

2. age, gender, are included as nodes in addition to glycans in the pearson correlation

computation. Why are these absent in the networks? Were these treated as confounders. While the other variables are continuous, I understand that gender is a categorical variable - was this factor in the partial correlation computation.

3. Transition from partial correlation to GGM: What GGM algorithm was used to arrive at the final structure? What are the underlying assumptions? Were the resulting networks weighted graphs?

4. Modularity and communities in the network abstraction: I understand the modularity was detected using the algorithm proposed by the last author (Krumhsiek) Is there any rationale behind the specific choice of this community structure detection algorithm? Does this algorithm permit overlapping communities?

5. What is degree preserving random edge rewiring? Were these used as internal controls in establishing statistical significance?

6. I am unclear as to how the associations with GWAS data from another population fortifies the claims in the manuscript.

7. Please summarize the primary contribution. It is still unclear to me, since you say that the actual pathway is not known. Without ground truth and absence of experimental manipulation I believe it might not be possible to justify or validate the findings with any rigor. I do agree some consensus has been shown between the croatian cohorts.

8. Since no raw or normalized data was made available in the supplementary documents, there is no way I could analyze them or validate the findings. I am afraid the journal audience would encounter similar issues as the Data Availability statement clearly states that the authors cannot make the data available due to restrictions. This diminishes the transparency and reproducibility of the present study.

Response to the reviewers

General comment: All reviewers, as well as the editor, were mainly concerned with the lack of experimental validation of our predictions. To address their comments, we performed three different sets of experiments.

First, we exposed a pool of 2AB-labeled IgG glycans to B4GalT1 and MGAT3 and compared UPLC spectra before and after exposure to the enzymes to identify changes in glycan abundances. For B4GalT1, we were able to observe changes in the concentrations of G1FS1 and G2FS1, which is one of the reactions included in our predicted rule G1. For MGAT3, we were unfortunately unable to see any changes in any glycan profile, including for the known substrates of the enzyme, probably due to interference of the fluorescence label in the enzymatic reactions. Therefore, we could neither prove nor disprove our second predicted rule N2.

Second, we tested one of our rejected reactions on a synthetic substrate, namely the addition of bisecting GlcNAc on galactosylated non-fucosylated glycans (rule N1). As positive control, we also considered a known substrate of the enzyme, namely the structure G0. While for the latter we observed complete conversion to G0N within 3 hours, for the former we did not see any changes even after a prolonged experiment time (48h), confirming that this reaction is not taking place.

The third series of experiments regarded the localization of the three enzymes involved in our predicted reactions (B4GalT1, ST6Gal1 and MGAT3) in two different cell lines using confocal microscopy. The results indicate that a substantial proportion of these enzymes is localized in the same Golgi compartments, supporting the thesis that our inferred reactions might happen *in vivo* (and contradicting the current state of the art of enzyme intra-Golgi localization).

In summary, we could prove that one of our inferred reactions occurs *in vitro*, and that one of the rejected rules is in fact not occurring. The other predicted rule seemed experimentally inaccessible. Moreover, we found the enzymes involved in the predicted rules to be co-localized in two different cell types, which supports the claim of such reactions being possible *in vivo*.

Details of the validation experiments can be found in the new result sections “Experimental validation by enzymatic assays” and “Enzyme colocalization experiment in cell lines” and in the Methods paragraphs “Validation of inferred reactions *in vitro*” and “Determination of enzyme colocalization cell lines”.

Reviewers' comments:

Reviewer #1 (Remarks to the Author):

The manuscript of is an awe-inspiring example of the the application of mathematical methods to “big” glycomics data. I should rightaway confess that I lack the background to fully assess the statistics and even less the genetic analyses. However, I am surprised about the biosynthetic “rules” postulated by the authors as they

all oppose the existing knowledge on the substrate specificity of the various glycosyltransferases.

Given the fact that bisecting GlcNAc is used to produce non-fucosylated IgG on a large scale, it is surprising to learn about a pathway where galactosylated, bisected glycans are fucosylated (rule F2).

The reactions in rule F2, which describes fucosylation of bisected glycans, were included as a possibility in the model for completeness, but were not selected by the statistical approach. Therefore, according to our results they are not likely to occur *in vivo*, in agreement with the reviewer's comment.

We added a sentence in the second paragraph of the result section "A rule-based approach predicts new enzymatic reactions" to make this point clearer to the reader.

"Since we followed an unbiased approach, this included reactions for which *in vitro* experiments showed evidence of inhibition, e.g. the addition of fucose to the G0N structure"

In the discussion, the authors argue with possibly outdated models of enzyme location. In my view, this is not the point. The crucial point rather is the substrate specificity of the enzymes involved. The relevant glycosyltransferases have all been purified or expressed and amply characterized. Notably, the related literature does not seem to be cited. Certainly, no Harry Schachter paper is found in the literature list.

The reviewer has a valid point here. In the new *in vitro* experiments included in the revised manuscript (see top of the response letter), we addressed both the predicted substrate specificity of the enzymes and their localization inside the Golgi stacks. Moreover, a new supplementary file has now been included (Supplemental File S1), which includes the primary literature from which each reaction in the known pathway was derived from. The relevant publications now also appear in the main manuscript.

So, while the statistical apparatus applied to the data sets (whose quality I will not question) may be of finest nature, and although the fact that the accepted biosynthetic pathways in fact do emerge as "thick" edges in the model I am left with the uneasy feeling that this lemon has been squeezed out a bit too much. If I am wrong, the authors should not have too much trouble to add some wet chemistry proving their new "rules" right. Then it would be a splendid work. Unless this is possible, it appears – excuse me – a bit as art for arts' sake.

We agree and refer the reviewer to the top of this response letter, which briefly describes the new experiments we performed to validate our findings.

Reviewer #2 (Remarks to the Author):

What follows is a review of the article entitled "Network inference from glycoproteomics data reveals new reactions in the IgG glycosylation pathway" by Benedetti et al and submitted to Nature Communications. This article describes a method that utilizes IgG1 Fc N-glycan composition to identify novel enzyme-catalyzed steps in glycan biosynthesis. The authors compared pathway analysis using a Pearson coefficient with a partial

correlation coefficient and noted the partial correlation coefficient provided a superior result. The authors correctly identify a shortcoming in the in vitro descriptions of the Golgi N-glycan remodeling biosynthesis pathway: all potential N-glycan substrates and products of each enzyme are not known, furthermore, all potential IgG Fc-conjugated N-glycan substrates and products of each enzyme are not known. It is the latter the authors address and their data will likely inform the former as reactions catalyzed by the three glycosyltransferases identified (Fut8, B4GalT1 and MGat3) affect many N-glycoproteins. In general the manuscript was well written and clear. The points were clearly described and the conclusions are interesting. The statistical analysis is largely well explained, which this reviewer finds unusual and appreciates with respect to this manuscript. Despite a fairly promising start with 22(?) new potential connections, the authors justifiably settle on only three novel biosynthetic routes.

The reviewer is correct in that our inference approach starts from 22 possible enzymatic reactions. We then grouped those 22 reactions into 6 rules, and our results suggest that two out of those six rules, namely rule G1 and N2, are part of the IgG glycosylation pathway.

To make this clearer in the manuscript, we modified two sentences in the result section “A rule-based approach can be used to predict new enzymatic reactions”, that now read:

“In this way, starting from 22 single potential new reactions, we defined six rules, as described in Figure 5A and Table 2. [...] In the selected pathway model from this analysis, rules G1 and N2 were added to the known pathway (Figure 5C), which resulted in the inclusion of eight new enzymatic steps in the IgG glycosylation pathway.”

It should be mentioned that these are not novel biosynthetic pathways, just an identification of a new substrate for a particular enzyme (and not ones that are particularly surprising). Unfortunately this limits impact of the manuscript.

We believe that first of all this is a matter of inherently ambiguous use of the word "pathway" in the entire field. We made sure that we consistently refer to "reaction steps", "pathway steps" or "substrate specificities", and avoid talking about new "pathways". Moreover, we hope that the limit in impact is now addressed by the new experiments.

One other aspect limited the enthusiasm of this reviewer for the manuscript as presented. The authors note on line 203 “To the best of our knowledge, it is not feasible to prove the predicted reactions experimentally under in vivo conditions.” That is probably correct. They neglect to mention that it is likely possible, with current technologies, to test the reactions in vitro. Such a proof of principle would represent a significant step towards validating the statistical methods.

We agree and again refer the reviewer to the top of this response letter, where we outline our new experimental results obtained for the revision of the manuscript.

Furthermore, such a measurement, with adequate controls, would provide evidence an enzymologist would find interesting, most notably: are these unpredicted substrates utilized with reasonable enzyme kinetics? It does not appear from the manuscript that the authors are able to deconstruct relative flux through each of the pathways, which would likewise also improve the manuscript and strengthen the impact. If so, these data should be discussed in a clear manner.

The reviewer is correct, with our approach we cannot infer kinetics. This would require a whole different set of statistical tools and experimental approaches. Moreover, we hope that the new experimental results (see above) at least increase the credibility of the novel pathway steps as such.

Minor notes:

Line 186: "We replicated this findings" should read "We replicated these findings"

We corrected the mistake.

Under what hypothetical circumstances could connections occur that do not represent an enzyme catalyzed reaction? Cross subclass connections are found and are physically meaningless, as noted. It would be valuable to discuss limitations of the technique more thoroughly.

We included a new paragraph in the Discussion to address further limitations of the statistical approach.

Recent evidence from the Lau and Cobb groups independently indicates that galactosylation and sialylation of IgG Fc is spatially decoupled. It would be worthwhile to consider that argument in the paragraph starting on line 239 that notes a physical connection between the B4GalT1 and St6Gal1 enzymes in the Golgi. Line 251: it is premature to suggest MGat3 and B4GalT1 are colocalized based on the network analysis data presented in the manuscript.

We performed a colocalization experiment in CoCa-2 and COS-7 cells for enzymes GalT1, MGAT3 and ST6Gal1. The results indicate that in both these cell types the three enzymes are substantially colocalized in the same Golgi compartments.

See also the summary of our validation experiments at the top of this response letter.

Unrelated humorous note: the "10001 Dalmatians biobank" is well named.

Reviewer #3 (Remarks to the Author):

Overview of the article:

The present study investigates data driven approaches for inferring IgG glycosylation pathway from large-scale mass spectrometry measurements. The authors argue that the current knowledge on glycosylation pathway is incomplete especially in in-vivo settings as in-vivo systems do not lend themselves for experimental manipulations. Subsequently, several computational strategies are presented in this regards by the authors. Pair-wise correlations, partial correlations, GGMs for identifying significant associations with a glycosylation pathway template using Fisher's exact tests, and correlating the findings to SNP data. The results are demonstrated on four croatian cohorts from different islands.

Specific Comments:

1. The authors should clear ambiguities related to the "benchmark pathway" since they compare their data driven approach to this benchmark. How did they arrive at the benchmarks? This question is especially pertinent since they critique existing in-vivo glycosylation pathways as incomplete and warranting the proposed data driven study.

The "benchmark pathway" was derived from literature. To make the information more transparent and accessible to the reader, we now created a new supplement (Supplemental File S1), where we provide the primary literature for each reaction of the known IgG glycosylation pathway.

2. Can glycosylation pathway vary across the four different croatian cohorts in the study? Is it important to have a whole genome sequencing across these cohorts to establish the association between potential sequence variations and its impact on the glycosylation?

The reviewer raises a very interesting point. There is no prior knowledge regarding population-specific IgG glycosylation pathways, so in this paper we assumed them to be the same for all cohorts. No genome sequencing data was available to us, but a large-scale glycan GWAS meta-analysis is currently being performed including eight IgG glycan cohorts. Results seem to indicate that, for SNPs in the genes coding for the four glycosyltransferases involved in IgG glycosylation, effects are in the same direction and of comparable size across the different populations.

At the end of this document (Figure 1) we attached few forest plots from this study, which we cannot cite in the manuscript at this point however, since the data is unpublished.

3. Is the glycosylation pathway a directed cyclic or acyclic graph? i.e. can it have feedback.

The pathway network is a directed, acyclic graph, as can be seen from Figure 2. We decided to not include this in the manuscript, since it is a mere depiction of the known reactions in a concise figure.

Of note, this touches a more complicated topic that we cannot address in this manuscript. While metabolic reaction systems can have cycles (e.g. the TCA cycle), actual "feedback" would rather refer to the regulation of the pathway, i.e. by end-product inhibition. Addressing this topic with the type of data we currently have does not seem feasible at this point.

Direct dependencies: Pair-wise correlations essentially models potential associations as undirected weighted graphs where weights may represent the magnitude of correlation. The authors have used pearson correlation in conjunction with FDR for modeling these. Fine.

Indirect Dependencies: On the other hand, dependencies between a pair of variables can be indirect and mediated through other variable(s) in which case one might model conditional dependencies using measures such as partial correlations as opposed to pairwise dependencies.

Questions and concerns:

1. If the objective is accommodate indirect dependencies then why are the results of Pearson correlation shown? Since dependencies modeled using partial correlation is a subset of those modeled using pearson correlation, it is well known that the corresponding graphs are less dense. So the figures on pearson correlation contribute no information and can be removed.

All of what the reviewer states is correct. Nevertheless, we believe that, since it is the first time that glycan correlations were investigated at a global level, it was important to show to the reader what the data correlation structure looks like. By showing how dense the Pearson correlation matrix is, even after multiple testing correction, we justify the introduction of a more sophisticated correlation measure to identify direct dependencies among the data. Therefore, we chose to keep the results on Pearson correlation.

2. age, gender, are included as nodes in addition to glycans in the pearson correlation computation. Why are these absent in the networks? Were these treated as confounders. While the other variables are continuous, I understand that gender is a categorical variable - was this factor in the partial correlation computation.

Age and gender were treated as confounders. Their effect was accounted for in the estimation of both types of correlation coefficients, but they were not included in the network representation, as neither of the variables can be assumed to be normally distributed (as the reviewer states). In fact, when calculating partial correlations, one can correct for non-normal variables, but one should not look at the "network links" these variables exhibit.

We clarified this in the Methods section by adding the sentence "This is done by including the confounding variables in the GGM calculation, but not showing them as nodes in the final network."

3. Transition from partial correlation to GGM: What GGM algorithm was used to arrive at the final structure? What are the underlying assumptions? Were the resulting networks weighted graphs?

First, we would like to clarify that a full-order (i.e. corrected for all other variables) partial correlation network is called a GGM, i.e. significant partial correlations and GGM are the same thing. We now clearly

state this in the methods section. To estimate the GGM, we used the GeneNet algorithm followed by multiple testing correction via Benjamini-Hochberg method (FDR 0.01), as reported in the Methods section of the manuscript. The resulting network is a weighted graph, which we now also state in the results.

The basic underlying assumption for any GGM computation is that data are normally distributed. For the data considered in this study, log-transformation was applied to improve normality prior to analysis. We modified the last sentence of the “Data preprocessing” paragraph in the Methods to clarify these steps.

“To verify normal distributions, we compared QQ-plots against normal distributions for non-logarithmized as well as logarithmized glycan data (Supplementary Figure 9). Since all distributions were closer to log-normality than to normality, we log-transformed the glycan concentrations prior to analysis.”

4. Modularity and communities in the network abstraction: I understand the modularity was detected using the algorithm proposed by the last author (Krumisiek) Is there any rationale behind the specific choice of this community structure detection algorithm? Does this algorithm permit overlapping communities?

The modularity calculation was adapted from the method introduced by Newman (Physical Review E, 2004), which was originally introduced as a clustering algorithm. In this paper, however, we wanted to compute the modularity of three predetermined clusters, identified by the three measured IgG subclasses. To make this more clear in the manuscript, we added the following sentence in the “Correlation networks and modularity” paragraph of the Methods.

“The network modularity algorithm was adapted from the widely used community detection clustering method of Newman³⁸, which optimizes the modularity Q to determine the nodes clusters. In this paper, we used the Q measure to assess the modularity of predefined clusters, defined by the three IgG subclasses.”

To make the primary contributions evident, we also modified a sentence in the result section, which now reads:

“To investigate this observation quantitatively, we calculated a subclass-based network modularity for all significantly positive edges based on the method by Newman et al (2004), and as previously adapted in Krumisiek et al (2011). We used degree-preserving random edge rewiring as a null-model to assess the statistical significance.”

Regarding overlapping communities: The original method does not allow to detect such clusters. However, we did not require this property from the method, since the analyzed subclasses are non-overlapping.

5. What is degree preserving random edge rewiring? Were these used as internal controls in establishing statistical significance?

Yes, the results of this procedure were used as an internal control to establish significance. The details of the degree preserving random edge rewiring have been now clarified in the Methods.

“To assess the significance of the observed modularity, we performed graph randomization via edge rewiring^{39,40}. In this process, two edges in the original data-driven network are randomly selected and the end nodes of each edge are swapped. The operation was repeated 10 times the number of edges to reach sufficient randomization. The entire randomization was repeated 10^5 times to obtain a sufficient number of null model networks.”

6. I am unclear as to how the associations with GWAS data from another population fortifies the claims in the manuscript.

Without the experiments of the original manuscript, GWAS was the closest we could get to a validation. By showing a link between SNPs in a glycosyltransferases gene region and a glycan product-over-substrate ratio of one of our predicted reactions, we strengthened our claim that that predicted reaction might happen *in vivo*.

Importantly, we would like to point out that the validation part in the manuscript is now substantially changed. We performed extensive experiments to confirm our predictions, please see top of this response letter. Thus, the GWAS is now not the endpoint of the story anymore, but rather a step towards the experimental validation experiments. We also changed the title of that section to and reformulated the text, in the light of the new experimental data. If required, we could move this part to the supplementary material. Moreover, we deleted two prominent GWAS paragraphs from the discussion, what would now distract from the true experimental validation.

7. Please summarize the primary contribution. It is still unclear to me, since you say that the actual pathway is not known. Without ground truth and absence of experimental manipulation I believe it might not be possible to justify or validate the findings with any rigor. I do agree some consensus has been shown between the croatian cohorts.

Regarding the ground truth: As mentioned above, to make the information more accessible to the reader, we now created a new supplement (Supplemental File S1) where for each reaction in the known IgG glycosylation pathway the primary literature is indicated.

Regarding the experiments: As mentioned above, we now performed extensive experiments to validate our findings, see top of the response letter.

8. Since no raw or normalized data was made available in the supplementary documents, there is no way I could analyze them or validate the findings. I am afraid the journal audience would encounter similar issues as the Data Availability statement clearly states that the authors cannot make the data available due to restrictions. This diminishes the transparency and reproducibility of the present study.

We do agree with the reviewer. Preprocessed glycomics data for the 4 discovery and replication cohorts are now available as a supplementary material.

Forest plots

[Redacted]

REVIEWERS' COMMENTS:

Reviewer #1 (Remarks to the Author):

I see that the authors have gone a long and very serious way to rebut all my points. Their answers and changes are entirely acknowledged. So, no further objections from my side.

Reviewer #2 (Remarks to the Author):

I am largely satisfied with the revisions to the manuscript. The authors increased the interest of the manuscript by performing the suggesting experiments and revising the text. Two small things to note:

1) The in vitro enzyme studies were performed with fluorophore-conjugated N-glycans, rather than IgG-conjugated N-glycans. I don't think this is a huge mis-step, however, the intent was to discover novel pathways for IgG-conjugated N-glycan biosynthesis. Perhaps the authors did attempt experiments with IgG-conjugated N-glycans and were unsuccessful? I would consider including a statement covering such experiments, if performed.

2). The results of the co-localization experiments were compelling, however, a small problem exists. One, plasma cells, and neither CaCo2 nor COS7 cells produce antibodies in the human body. These cell lines may or may not appropriately recapitulate Golgi conditions found in plasma cells. Second, the resolution may not be sufficient to demonstrate actual catalytically-relevant colocalization. I cannot suggest an alternative approach, and I believe the authors interpreted the results appropriately.

Response to the reviewers

Reviewers' comments:

Reviewer #1 (Remarks to the Author):

I see that the authors have gone a long and very serious way to rebut all my points. Their answers and changes are entirely acknowledged. So, no further objections from my side.

We thank the reviewer for the suggestions and remarks.

Reviewer #2 (Remarks to the Author):

I am largely satisfied with the revisions to the manuscript. The authors increased the interest of the manuscript by performing the suggesting experiments and revising the text. Two small things to note:

1) The in vitro enzyme studies were performed with fluorophore-conjugated N-glycans, rather than IgG-conjugated N-glycans. I don't think this is a huge mis-step, however, the intent was to discover novel pathways for IgG-conjugated N-glycan biosynthesis. Perhaps the authors did attempt experiments with IgG-conjugated N-glycans and were unsuccessful? I would consider including a statement covering such experiments, if performed.

We only performed the experiments outlined in the manuscript. No additional attempts to experimentally prove the inferred reactions were made.

2) The results of the co-localization experiments were compelling, however, a small problem exists. One, plasma cells, and neither CaCo2 nor COS7 cells produce antibodies in the human body. These cell lines may or may not appropriately recapitulate Golgi conditions found in plasma cells.

The reviewer raises an important point: Unfortunately, current technologies do not allow to perform glycosyltransferase localization experiments on plasma cells due to their small size (only around 7 μm in diameter). We do agree that a final prove of the enzyme colocalization should have been carried out in plasma cells, but we are also convinced that the replication of the results in two different cell lines is a good indicator that the enzymes colocalize in other cell types.

Second, the resolution may not be sufficient to demonstrate actual catalytically-relevant colocalization. I cannot suggest an alternative approach, and I believe the authors interpreted the results appropriately.

The results that we show in the manuscript are based on the available equipment and could thus not produce images at higher resolution. Considering the compactness of the organelle membranes in COS-7 cells, even with higher resolution we would expect to obtain the same result with slightly less variation. That is also the main reason why we considered CaCo-2 cells for comparison, as the individual Golgi stacks are more visible.

One way to improve the results would be to use immunogold labelling of enzymes with EM to have a better view of distribution in the membranes, but we do not have proper antibodies available for this kind of assay.